# Learning on Random Balls is Sufficient for Estimating (Some) Graph Parameters

**Takanori Maehara**
Facebook AI
London, United Kingdom
tmaehara@fb.com

**Hoang NT**
Tokyo Tech & RIKEN AIP
Tokyo, Japan
hoangnt@net.c.titech.ac.jp

## Abstract

Theoretical analyses for graph learning methods often assume a complete observation of the input graph. Such an assumption might not be useful for handling any-size graphs due to the scalability issues in practice. In this work, we develop a theoretical framework for graph classification problems in the partial observation setting (i.e., subgraph samplings). Equipped with insights from graph limit theory, we propose a new graph classification model that works on a randomly sampled subgraph and a novel topology to characterize the representability of the model. Our theoretical framework contributes a theoretical validation of mini-batch learning on graphs and leads to new learning-theoretic results on generalization bounds as well as size-generalizability without assumptions on the input.

## 1 Introduction

Going beyond regular structural inputs such as grids (images), sequences (time series, sentences), or general feature vectors is an important research direction of machine learning and computational sciences. Arguably, most interesting objects and problems in nature can be described as graphs [37]. For such reason, graph learning methods, especially Graph Neural Networks (GNN) [60], have recently proven to be a useful solution to many problems in computer vision [10, 12, 24, 63], complex network analyses [22, 30, 73], molecule modeling [17, 32, 41, 45], and physics simulations [3, 31, 54].

The significant value of graph learning models in practice has inspired a large amount of theoretical work dedicated to exploring their representational limits and the possibilities of improving them. Most notably, the representational capability of GNNs has been in the spotlight of recent years. To answer the question "*Can GNNs approximate all functions on graphs?*", researchers discussed universal invariant and equivariant neural networks [27, 40, 43, 51] as *theoretical* upper limits for neural architectures or showed the correspondence between message-passing GNNs (MP-GNNs) to the Weisfeiler-Leman (WL) algorithm [68] as *practical* upper limits [46, 70].

Given an extremely large graph as an input, it is often impractical to keep the whole graph in the working memory. Therefore, practical graph learning methods often utilize neighborhood samplings [22, 73] or random walks [53] to handle this scalability issue. Because existing analyses assumed a complete observation of the input graphs [27, 51, 57], it is unclear what can be learned if we combine graph learning models with random samplings. Thus, the relevant question in this scenario is "*What graph functions are representable by GNNs when we can only observe random neighborhoods?*" This question adds another dimension to the discussion of GNN expressivity; even if we have a powerful GNN (in both theoretical and practical senses), what kind of graph functions can we learn if the input graphs are too large to be computed as a whole?

**Contributions** This study proposes a theoretical approach to address graph learning problems on large graphs by identifying a novel topology of the graph space. We discuss the graph classification

problem in the main part of the paper and extend the discussion to the vertex classification problem in Appendix C. The extension to vertex classification can be realised by viewing it as a rooted-graph classification problem. We first introduce a *random ball sampling GNN (RBS-GNN)*, which is a mathematical model of GNNs implementable in a *random neighborhood* computational model, and prove that the model is universal in the class of estimable functions (Theorem 4). Our main contribution is introducing *randomized Benjamini–Schramm topology* in the space of all graphs and identifying the estimability of the function as the uniform continuity in this topology (Theorem 7). By applying our main theorem, we obtain the following learning-theoretic results.

- We show the equivalent of estimability and continuity (Theorems 4 and 7). This implies the continuity assumption is a sufficient condition for the mini-batch learning on graphs.
- We prove that the functions representable by RBS-GNNs are generalizable by showing an upper bound of the Rademacher complexity of Lipschitz graph functions (Theorem 10).
- We identify size-generalizable functions with estimable functions (Theorem 11). Then, by recognizing the size-generalization as a domain adaptation, we provide a size-generalization error based on the Wasserstein distance (Theorem 13).

Unlike existing studies, which assumed a random graph model [26, 28] or boundedness [11, 27, 51, 58], our framework does not assume anything about the graph class; instead, we assume the continuity of the graph functions. Our results listed above are model-agnostic, i.e., we only discuss the property of the function space, regardless of how GNN models are implemented. The model-agnostic nature of our results gives a systematic view to general graph parameters learning; their generality is especially useful as there are many different GNN architectures in practice [57, 69, 78].

## 2  Related Work

**Large-scale GNNs**  The success of GNNs, especially vertex classification models like GCN [30] and GraphSAGE [22], has led to various large-scale industrial GNN systems (see [1] and references therein). Aiming to increase computational throughput while maintaining the predictive performance, most of these systems implemented fixed-size neighborhood sampling [22] to enable large-scale batching [23, 73, 76, 77]. GNNs have also been applied to the 3D point clouds classification problem [67], which translates a computer vision problem to the large graph classification problem, and the random sampling was empirically shown to be effective [35]. In this context, our work contributes a theoretical justification for the random sampling procedure.

**Graph Parameter Learning**  Graph function, graph parameter, or graph invariant refer to a (real or integer value) property of graphs, which only depends on the graph structure. In other words, they are functions defined on isomorphism classes of graphs [37]. Determining graph properties from data has long been a topic of interest in theoretical computer science [8, 37] and is an important machine learning task in computational chemistry [9, 17] and biology [6, 16]. Recently, GNNs have been proven successful on a wide range of graph learning benchmark datasets. Current literature analyzed their expressivity to gain a better understanding of the architectures [27, 44, 51]. Several works identify MP-GNNs to the 1-dimensional WL isomorphism test [70] and further improve the GNN architectures to more expressive variants such as $k$-dimensional WL [46], port-numbered message passing [58], and sparse WL [48]. GNNs are also linked to the representational power of logical expressions [2]. These theoretical results assumed the complete observation of the input graph; therefore, it is difficult to see to what extent these results would hold when the only partial observation is available. By studying the RBS-GNN model, we give an answer to this issue. We use GNNs because they are the most expressive graph learning methods [27, 70]. Nonetheless, our results generalize for other universal (Theorem 4) and partially-universal (Theorem 17) methods.

**Generalization**  Besides expressivity, another challenge in graph learning is to understand the generalization bounds. Scarselli et al. [61] introduced an upper bound for the VC-dimension of functions computable by GNNs, in which the output is defined on a special supervised vertex. Garg et al. [15] derived tighter Rademacher complexity bounds for similar MP-GNNs by considering the local computational tree structures. Liao et al. [36] obtained a generalization gap of MP-GNNs and GCNs [30] using PAC-Bayes techniques. Du et al. [11] obtained a sample complexity using a result in the kernel method for their graph neural tangent kernel model in learning propagation-based functions. Verma and Zhang [66] obtained a generalization gap of single-layer GCNs by analyzing the stability and dependency on the largest eigenvalue of the graph; Lv [39] derived a Rademacher bound for a similar

GCN model with a similar dependency. Keriven et al. [28] assumed an underlying random kernel (similar to graphons [37]) and analyzed the stability of discrete GCN using a continuous counterpart c-GCN. They derived the convergence bounds by looking at stability when diffeomorphisms [42] are applied to the underlying graph kernel, the distribution, and the signals. All these methods placed some assumptions on the graph space; either bounded degree [15, 36, 39], bounded number of vertices [11, 61, 66], or graphs belong to a random model [28]. Therefore, all these results become either inapplicable or unbounded in the general graph space. Our Theorem 10 contributes a complexity bound without assumptions on the graphs.

**Property Testing and Constant-Time Local Algorithms** Property testing on graphs is a task to identify whether the input graph satisfies a graph property $\Pi$ or $\epsilon$-far from $\Pi$ [18]. Often a researcher in this area tries to derive an algorithm whose complexity is constant (i.e., only depends on $\epsilon$) or sublinear in the input size [56]. Several graph properties admit sub-linear (or constant-time) algorithms; the examples include bipartite testing, triangle-free testing, edge connectivity, and matching [49, 74]. Recently, by bridging the constant-time algorithms and the GNN literature, Sato et al. [59] showed that, for each vertex, the neighborhood aggregation procedure of a GNN layer (they called it "node embedding") can be approximated in constant time. However, this does not result in a constant-time learning algorithm for GNNs because we still need to access all the vertices to get the desired outputs. Our results provide the first "fully constant-time" GNNs in the sense that the whole learning and prediction process runs in time independent of the size of the graphs (Section 4).

**Statistical Theory for Network Analysis** Learning graph property from samples is a traditional topic in statistical network analysis [14, 21, 34]. Klusowski and Wu [33] proved that it is hard to estimate the number of connected components using sublinear-size samples. Bhattacharya et al. [5] showed that the Horvitz–Thompson estimator for the number of subgraphs of constant size is consistent and asymptotic normal if the fourth-moment condition holds. Our result is consistent with these results as the number of connected components is non-continuous and the number of motifs is continuous in the randomized Benjamini–Schramm topology. These studies indicate a future direction of this study; for example, the asymptotic normality and consistency of RBS-GNNs.

## 3  Preliminaries

### 3.1  Graphs

A *(directed) graph* $G$ is a tuple $(V, E)$ of the set of vertices $V$ and the set of edges $E \subseteq V \times V$. We use $V(G)$ for $V$ and $E(G)$ for $E$ when the graph is unclear from the context. A graph is *weakly connected* if the underlying undirected graph has a path between any two vertices. A *weakly connected component* is a maximal weakly connected subgraph. Two graphs $G$ and $H$ are *isomorphic* if there is a bijection $\phi : V(G) \to V(H)$ such that $(\phi(u), \phi(v)) \in E(H)$ if and only if $(u, v) \in E(G)$. Let $\mathcal{G}$ be the set of all directed graphs. A *ball of radius $r$ centered at $v$*, $B_r(v)$ (also simply $B$), is the set of vertices whose shortest path distance from $v$ is bounded by $r$. For $U \subseteq V(G)$, $G[U]$ is the subgraph of $G$ induced by $U$.

A *rooted graph* $(G, v)$ is a graph $G$ augmented with a vertex $v$ in $V(G)$. The isomorphism between $(G, v)$ and $(H, u)$ is defined in the same way as for graphs with the extra requirement that it maps $v$ to $u$. A *$k$-rooted graph* $(G, v_1, \ldots, v_k)$ is defined similarly. We often recognize the graph induced by the ball $B_r(v)$ as a rooted graph whose root is $v$ and by the union of $k$ balls as a $k$-rooted graph.

Modern graph learning problems ask for a function $p : \mathcal{G} \to \mathcal{D}$ from training data, where $\mathcal{D}$ is a "learning-friendly" domain such as the set of real numbers $\mathbb{R}$, a $d$-dimensional real vector space $Q \subseteq \mathbb{R}^d$, or some finite sets. In most cases, the function $p$ is required to be isomorphism-invariant (or invariant for short). This notion of graph functions coincides with the definition of *graph parameters*. Another term used in the literature is *graph property*, which can be formalized as a graph function whose co-domain is $\{0, 1\}$. Our work focuses on the case in which the co-domain is $\mathbb{R}$.

### 3.2  Computational Model

Extremely large graphs are usually stored in some complicated storage. Thus, there are some constraints on how we can access the graphs. In the area of property testing, such a situation is modeled by introducing a *computational model*, which is an oracle for accessing the graph.

Importantly, each computational model induces a topology on the graph space. As we will show in later sections, the ability to represent graph functions is related to this topology.

There are three main computational models in the literature: the *adjacency predicate model* [20], the *incidence function model* [19], and the *general graph model* [25, 52]. The adjacency predicate model, also known as the dense graph model, allows randomized algorithms to query whether two vertices are adjacent or not. With the incidence function model, also known as the bounded-degree graph model, algorithms can query a specific neighbor of a vertex. The general graph model lets the algorithms ask for both a specific neighbor and for whether two vertices are adjacent; hence, this is the most realistic model for actual algorithmic applications [18].

In this study, we consider the following *random neighborhood model*, which allows us to access the input graph $G$ via the following queries:

- SampleVertex($G$): Sample a vertex $u \in V$ uniformly randomly.
- SampleNeighbor($G$, $u$): Sample a vertex $v$ from the neighborhood of $u$ uniformly randomly, where $u$ is an already obtained vertex.
- IsAdjacent($G$, $u$, $v$): Return whether the vertices $u$ and $v$ are adjacent, where $u$ and $v$ are already obtained vertices.

This model is a randomized version of the general graph model. Czumaj et al. [8] proposed a similar model to analyze edge streaming algorithms for property testing. However, their model does not have the IsAdjacent query, i.e., it is a randomized version of the incidence function model. Note that the computational model naturally specifies the *estimability* of the graph parameters. More formally, we have the following definition of estimability with respect to our random neighborhood model.

**Definition 1** (Constant-Time Estimable Graph Parameter). *A graph parameter $p$ is constant-time estimable on the random neighborhood model (estimable for short) if for any $\epsilon > 0$ there exists an integer $N$ and a randomized algorithm $\mathcal{A}$ in the random neighborhood model such that $\mathcal{A}$ performs at most $N$ queries and $|\mathcal{A}(G) - p(G)| < \epsilon$ with probability at least $1 - \epsilon$ for all graphs $G \in \mathcal{G}$.*

Some examples of (non-)estimable graph parameters are:

**Example 2.** *The number of vertices, min/max degree, and connectivity are not estimable.*

**Example 3.** *The triangle density and the local clustering coefficient are estimable.*

Additional examples of estimable graph parameters and experimental results are provided in Appendix D. In the next section, we implement a GNN following the proposed random neighborhood computational model. By showing the connection between the GNN and algorithms in the random neighborhood model, we obtain several theoretical results in Section 5.

## 4 Random Balls Sampling Graph Neural Networks (RBS-GNN)

This section introduces *RBS-GNN*, a theoretical GNN architecture based on the random neighborhood model. RBS stands for "Random Balls Sampling" and also "Random Benjamini–Schramm" because our random neighborhood model extends the topology of the Benjamini–Schramm convergence [4]. Given an input graph, an RBS-GNN samples $k$ random vertices and proceeds to sample random balls $B_1, \ldots, B_k$ rooted at each of these vertices. A random ball of radius $r$ and branching factor $b$ is a subgraph obtained by the procedure RandomBallSample, illustrated in Figure 1 for $r = 1$ and $b = 4$. The exact procedure is presented in Algorithm 1. It is trivial to see that RandomBallSample can be implemented under the random neighborhood model with SampleVertex and SampleNeighbor.

After sampling $k$ random balls, the next step is identifying the induced subgraph $G[B_1 \cup \cdots \cup B_k]$ using the IsAdjacent procedure and computing the weakly connected components $C_1, \ldots, C_{N_C}$ of the induced subgraph (Step 3 of Figure 1). The classifier part of an RBS-GNN has two trainable components: a multi-layer perceptron $g$ and a GNN $f$. The output of an RBS-GNN is defined as

$$\text{RBS-GNN}(G) = g\left(\sum_j f(C_j)\right). \tag{1}$$

It should be emphasized that, as mentioned in the end of Section 2, our RBS-GNN can be evaluated in constant time (i.e., only dependent on the hyperparameters) because the subsets returned by RandomBallSample has a constant size regardless of the size of input graph $G$.

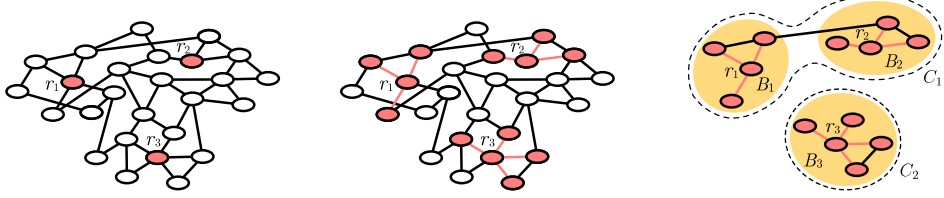

Step 1: Sample random roots ▶ Step 2: Sample random neighbors ▶ Step 3: Get induced subgraphs and connected components

Figure 1: Random Balls Sampling Procedure (Algorithm 1). Our computational model is different from the existing general graph model at Step 2, where we sample neighbors randomly instead of taking all neighbors. In Step 3, the randomly sampled edges are shown with color, and the induced edges are black. The weakly connected components $C_1$ and $C_2$ are inputs to the GNN.

---

**Algorithm 1** Randomized Benjamini–Schramm GNN

---

1: **procedure** RANDOMBALLSAMPLE($G, b, r$)
2:     $\texttt{layer}[0] \leftarrow [], \ldots, \texttt{layer}[r] \leftarrow []$
3:     Sample one random vertex from $V(G)$ and insert to $\texttt{layer}[0]$
4:     **for** $i = 1, \ldots, r$ **do**
5:         **for** $u$ in $\texttt{layer}[i-1]$ **do**
6:             Sample $b$ random vertices (with replacement) from $\mathcal{N}(u)$ and insert to $\texttt{layer}[i]$
7:     **return** $G[\texttt{layer}[0] \cup \cdots \cup \texttt{layer}[r]]$
8: **procedure** RBS-GNN($G, f, g, b, r, k$)
9:     $B_1, \ldots, B_k \leftarrow \textsf{RandomBallSample}(G, b, r)$          ▷ Runs $k$ times to get $k$ balls.
10:     $C_1, \ldots, C_{N_C} \leftarrow \textsf{WeaklyConnectedComponents}(G[B_1 \cup \ldots B_k])$
11:     **return** $g(\sum_j f(C_j))$

---

**Relation to Existing GNN Models** While RBS-GNN is motivated by the random neighborhood model, it has a strong connection with existing message-passing GNNs and optimization techniques in graph learning. When we select $f$ to be a simple message-passing GNN, RBS-GNN is a generalization of the mini-batch version of GraphSAGE (Algorithm 2 in [22]), and the multi-layers perceptron module $g$ acts as the global READOUT as in the GIN [70] architecture. Our analysis technique also applies for other mini-batch approaches of popular graph learning models [30, 64, 65]. This result provides a theoretical validation for the mini-batch approach in practice. Note that this is a positive result for continuous graph parameters, and we make no claim for the non-continuous case. On the other hand, $f$ can also be a more expressive variant such as high-order WL [48], $k$-treewidth homomorphism density, or a universal approximator [27, 51].

## 5 Main Result

In this section, we conduct theoretical analyses of RBS-GNN for the graph classification problem. All the proofs are in Appendix A. To simplify the analysis, we assume the hyperparameters $k$, $b$, and $r$ have the same value, and by a slight abuse of notation, we denote these values by $r$. Note that this setting would not alter the notion of estimability. We further simplify the discussion by assuming the graphs have no vertex features. Similar results hold when the vertices have finite-dimensional vertex features; see Appendix B. Additionally, as mentioned in Section 1, we obtained complementary results for the vertex classification problem in Appendix C.

### 5.1 Universality of RBS-GNN

We first characterize the expressive power of RBS-GNN. The following shows the universality of RBS-GNN, with a universal GNN component $f$, in the space of the estimable functions.

**Theorem 4** (Universality of RBS-GNN). *If a graph parameter $p : \mathcal{G} \rightarrow \mathbb{R}$ is estimable (in the random neighborhood model), then it is estimable by an RBS-GNN with a universal GNN $f$.*

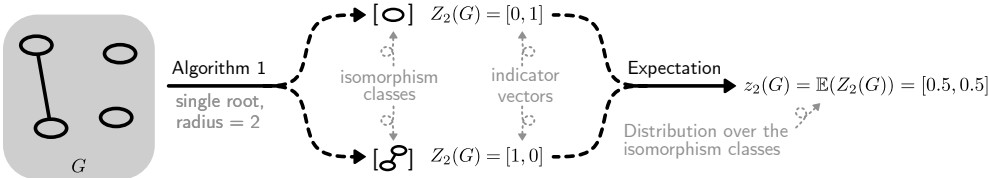

Figure 2: An illustrating example for the $r$-profile with $k = 1, r = 2$, and a simple graph $G$.

The proof of this theorem is an adaptation of the proof techniques by Czumaj et al. [8]. We first introduce a *canonical estimator*, which is an algorithm in the random neighborhood model defined by the following procedure. (1) Sample $r$ random balls $B_1, \ldots, B_r$ using RandomBallSample($G$, $r$, $r$); (2) Return a number according to the isomorphism class of the subgraph $G[B_1 \cup \cdots \cup B_r]$ induced by the balls. Since the number of random balls, the branching factor, and the radius are constant, we can see that the size of $G[B_1 \cup \cdots \cup B_r]$ is bounded by $r^{r+2}$. Therefore, we can list all isomorphism classes of all graphs having at most $r^{r+2}$ vertices and assign a unique number to each of them. Also, since the induced subgraph is bounded, it is possible to construct a universal approximator GNN [27, 51]. Therefore, we obtained the following.

**Lemma 5.** *If a graph parameter $p$ is estimable, then it is estimable by a canonical estimator.*

Since a canonical estimator assigns a number according to the isomorphism class of the input, we see that RBS-GNN can approximate the canonical estimator by letting $f$ be a universal approximator for bounded graphs. See the proof in Appendix A.1 for more detail.

**Relation to Universality Results** Existing universal GNNs assumed that the number of vertices of the input graphs are bounded. Theorem 4 shows that these universal GNNs for bounded graphs can be extended to general graphs by approximating the general graphs using the random balls sampling procedure. As a drawback, the theorem is only applicable to the continuous functions in the randomized Benjamini–Schramm topology introduced below. We emphasize that this drawback shows the limitation of the partial-observation (random neighborhoods) setting.

### 5.2 Topology of Graph Space: Estimability is Uniform Continuity

The previous section defined the estimability by the existence of an estimation algorithm. Such definition is suitable for algorithmic analysis; however, it is not suitable for further analysis, such as deriving the generalization error bounds. This section rephrases our estimability by the continuity in a new topology induced by a distance between two graphs.

We start with a simple example in Figure 2. The figure shows an input graph $G$ consisting of two isolated vertices and one edge. For simplicity, Algorithm 1 only samples a single ball ($k = 1$) with radius two ($r = 2$). This configuration let us obtain two isomorphism classes with equal probability: a single vertex and a single edge. Hence, the event which each isomorphism class is obtained from Algorithm 1 can be represented by a two-dimensional vector. We denote this vector $Z_2(G)$, which takes value $[0, 1]$ when the single vertex is sampled and value $[1, 0]$ when the single edge is sampled. Clearly, this is a random vector whose expectation defines a distribution over the isomorphism classes with respect to the configuration of Algorithm 1. More generally, we can define the $r$-profile $Z_r(G)$ and the corresponding isomorphism class distribution $z_r(G)$ for positive integer values of radius $r$.

For an integer $r$, an *$r$-profile* $Z_r(G)$ of a graph $G$ is a random variable of the ($k$-rooted[1]) isomorphism class of $G[B_1 \cup \cdots \cup B_r]$, where each $B_j$ is obtained from RandomBallSample($G$, $r$, $r$). As RandomBallSample($G$, $r$, $r$) produces a graph of size at most $r^r$, we can identify $Z_r(G)$ as a random finite-dimensional vector. Let $z_r(G) = \mathbb{E}[Z_r(G)]$ be the probability distribution over the isomorphism classes in terms of the $k$-rooted graph isomorphism, where the expectation is taken over SampleVertex and SampleNeighbor. The *sampling distance* of two graphs is defined by

$$d(G, H) = \sum_{r=1}^{\infty} 2^{-r} d_{TV}(z_r(G), z_r(H)), \tag{2}$$

---

[1]For simplicity, we let $k = r$.

where $d_{TV}$ is the total variation distance of two probability distributions given by $d_{TV}(p, q) = (1/2)\|p - q\|_1$. It should be emphasized that the sampling distance allows us to compare any two graphs even though they have a different number of vertices. We call the topology on the set of all graphs $\mathcal{G}$ induced by this sampling distance *randomized Benjamini–Schramm topology*. A graph parameter is defined to be *uniformly continuous in the randomized Benjamini–Schramm topology* by the followings.

**Definition 6** (Uniform Continuity in the RBS Topology). *A graph parameter $p : \mathcal{G} \to \mathbb{R}$ (resp. a randomized algorithm $\mathcal{A}$) is uniformly continuous if for any $\epsilon > 0$ there exists $\delta > 0$ such that for any $G$ and $H$, $d(G, H) \leq \delta$ implies $|p(G) - p(H)| < \epsilon$ (resp. $|\mathcal{A}(G) - \mathcal{A}(H)| \leq \epsilon$ with probability at least $1 - \epsilon$).*

This topology connects the estimability in terms of the continuity as follows.

**Theorem 7.** *A graph parameter $p$ is estimable in the random neighborhood model if and only if it is uniformly continuous in the randomized Benjamini–Schramm topology.*

The "if" direction of this theorem is given by the triangle inequality and the (optimal) coupling theorem [7]; the "only-if" is proved using the fact that the graph space is totally bounded as follows.

**Lemma 8** (Totally Boundedness of Graph Space). *For any $\epsilon > 0$, there exists a set of graphs $\{H_1, \ldots, H_C\}$ with $C \leq 2^{2^{(\log 1/\epsilon)^{O(\log 1/\epsilon)}}}$ such that $\min_{j \in \{1, \ldots, C\}} d(G, H_j) \leq \epsilon$ for all $G$.*

Theorem 7 allows us to apply existing "functional analysis techniques" to analyze the estimable functions. We present such applications in Section 6.

**Intuition of the Sampling Distance and Relation to Benjamini–Schramm Topology** The intuition behind our definition of the sampling distance reflects the idea that two graphs are similar when random samples from these graphs look similar, where the similarities on small samples are more important to the similarities on large samples. This definition generalizes the Benjamini–Schramm topology for the space $\mathcal{G}_D$ of all graphs of degree bounded by $D$, which uses a different definition of the "$r$-profile:" Let us define the $r$-profile by the union of $r$ balls of radius $r$ whose centers are sampled randomly. Then, the sampling distance defined using this $r$-profile induces a topology called the *Benjamini–Schramm topology*. This topology was first studied by Benjamini and Schramm [4] to analyze the planar packing problem, and now it is widely used to analyze the limit of bounded degree graphs, where the limit object is identified as the graphing; see [38]. A practical issue of the Benjamini–Schramm topology is that it is only applicable to bounded degree graphs, where many real-world extremely large graphs are complex networks having power-law degree distributions (i.e., unbounded degree). We addressed this issue by introducing the randomized Benjamini–Schramm topology, which is applicable to all graphs.

## 6 Theoretical Applications

### 6.1 Robustness Against Perturbation

The continuity immediately implies the robustness against the structural perturbation, i.e., for any $\epsilon > 0$ there exists $\delta > 0$ such that the output of RBS-GNN does not change more than $\epsilon$ if the graph is perturbed at most $\delta$ in the sampling distance. As the perturbation in sampling distance may not be intuitive in practice, we here provide a bound regarding the additive perturbation edges.

**Proposition 9.** *Let $G$ be a graph and let $G'$ be the graph obtained from $G$ by adding $\delta|V(G)|$ edges completely randomly where $0 < \delta < 1$. Then $d(G, G') = O(1/\log(1/\delta))$.*

This result indicates that to change the output of RBS-GNN, one needs to add linearly many random edges; it is impractical in extremely large graphs. Note that the "adversarial" perturbation can change the distance more easily, especially if there is a "hub" in the graph; see Appendix A.3 for details.

### 6.2 Rademacher Complexity

Thus far, we only discussed the expressibility of the functions regardless of the learnability. Here, we derive the Rademacher complexity for the class of Lipschitz functions in the random Benjamini–Schramm topology. This gives an algorithm-independent bound of the learnability of the functions.

**Theorem 10.** *Let $n$ be the number of training instances. The Rademacher complexity $R_n$ of the set of 1-Lipschitz functions that maps to $[0, 1]$ is $(\log \log n)^{-O(1/\log \log \log \log n)}$. It is $o(1/\log \log \log n)$.*

This result implies that, by minimizing the empirical error of $n$ instances, we can achieve the generalization gap of $o(1/\log \log \log n)$ with high probability. To the extent of our knowledge, this is the first Rademacher bound for the general graph space, which guarantees the asymptotic convergence on any graph learning problem without assuming any graph structure.

**Comparison with Existing Results** The significant difference between existing studies [11, 15, 36, 39, 61, 66] and our bound (Theorem 10) is that ours is independent of any structural property, such as the maximum number of vertices, the maximum degree, and the spectrum of the graphs. Thus, ours can be applied to any graph distribution. Simply put, this is a consequence of the totally boundedness of the graph space (Lemma 8): For any $\epsilon > 0$, the space of all graphs is approximated by finitely many graphs; hence any graph parameter is bounded by the values among them, which is a constant depending on $\epsilon$. The drawback of this generality is its poor dependency on the number of instances $n$, which leaves significant room for quantitative improvement. One possible way to improve the bound is by assuming some properties of the graph distribution because the above derivation is distribution-agnostic; a concrete strategy for improvement is left for future works.

### 6.3 Size-Generalizability

One interesting topic of GNNs is *size-generalization*, which is a property that a model trained on small graphs should perform well on larger graphs. Size-generalization is observed in several tasks [29]; however, it has also been proved that some classes of GNNs do not naturally generalize [72]. Hence, we want to know about the conditions for GNNs to generalize.

We need to distinguish the "approximation-theoretic" size-generalizability and the "learning-theoretic" size-generalizability. The former is the possibility of size-generalization, which is proved by showing the existence of size-generalizing models. This, however, does not mean that a size-generalizable model is obtained by training; thus, we need to introduce the latter. The latter is the degree of size-generalizability when we train a model using a dataset (or a distribution); it is proved by bounding the generalization error.

#### 6.3.1 Approximation-Theoretic Size-Generalizability

We say that a function $p$ is *size-generalizable in approximation-theoretic sense* if for any $\epsilon > 0$, there exists $N > 0$ such that we can construct an algorithm $\mathcal{A}$ using dataset $\{(p(G_{\leq N}), G_{\leq N}) : |V(G_{\leq N})| \leq N\}$ such that $|p(G) - \mathcal{A}(G)| \leq \epsilon$ with probability at least $1 - \epsilon$ for all $G \in \mathcal{G}$. This gives one mathematical formulation of the size-generalizability as it requires to fit algorithm $\mathcal{A}$ to all graphs using the dataset of bounded graphs. In this definition, we have the following theorem.

**Theorem 11.** *Estimable functions are size-generalizable in the approximation-theoretic sense.*

This theorem is proved by constructing a size-generalizable algorithm. We first pick the continuity constant $\delta$ for $\epsilon$ using Theorem 7. Then, we construct a $\delta$-net using Lemma 8. By storing all the values $p(G_i)$ for the graphs in the $\delta$-net, we obtain a size-generalizable algorithm, where $N$ is the maximum number of the vertices in the $\delta$-net.

#### 6.3.2 Learning-Theoretic Size-Generalizability

From the learning theoretic viewpoint, size-generalization is a domain adaptation from the distribution of smaller graphs to the distribution of larger graphs [72]. Thus, it is natural to utilize the domain adaptation theory [55]. Especially since we have introduced the sampling distance defined on all pairs of graphs irrelevant to their sizes, we here employ the Wasserstein distance-based approach [62].

We start from a general situation. Let $\mathcal{D}_1$ and $\mathcal{D}_2$ be joint distributions of graphs and their labels, and $\mathcal{G}_1$ and $\mathcal{G}_2$ be the corresponding marginal distributions of graphs. We abbreviate $\mathbb{E}_1$ and $\mathbb{E}_2$ for the expectations on $\mathcal{D}_1$ and $\mathcal{D}_2$, respectively. The Wasserstein distance between $\mathcal{G}_1$ and $\mathcal{G}_2$ is given by

$$W(\mathcal{G}_1, \mathcal{G}_2) = \inf_{\pi} \mathbb{E}_{(G_1, G_2) \sim \pi} d(G_1, G_2), \tag{3}$$

where $d$ is the sampling distance of the graphs, and $\pi$ runs over the couplings between these distributions. Let $\lambda = \inf_h \{\mathbb{E}_1 |y - h(G)| + \mathbb{E}_2 |y - h(G)|\}$ be the optimal combined error, where $\inf_h$ runs over all 1-Lipschitz functions. We have the following lemma.

**Lemma 12.** *For any* 1-*Lipschitz functions* $h$ *and* $h'$, *we have the following.*

$$\mathbb{E}_1 |y - f(G)| \leq \mathbb{E}_2 |y - f(G)| + 2W(\mathcal{G}_1, \mathcal{G}_2) + \lambda, \tag{4}$$

Combining this result with the Rademacher complexity (Theorem 10), we obtain the following generalization bound.

**Theorem 13.** *Let* $\epsilon > 0$. *Let* $(y_{21}, G_{21}), \ldots, (y_{2n}, G_{2n})$ *be independently drawn from* $\mathcal{D}_2$. *If* $\lambda = O(\epsilon)$ *and* $n \geq 2^{2^{2^{\tilde{\Omega}(1/\epsilon)}}}$, *then, for any* 1-*Lipschitz function* $h$, *we have*

$$\mathbb{E}_{(G_1, y_1) \sim \mathcal{D}_1}[|y_1 - h_1(G)|] \leq \frac{1}{n} \sum_{i=1}^{n} |y_{2i} - h(G_{2i})| + 2W(\mathcal{D}_1, \mathcal{D}_2) + O(\epsilon) \tag{5}$$

*with probability at least* $1 - \epsilon$.

The condition $\lambda = O(\epsilon)$ requires the existence of a "consistent rule" among both $\mathcal{D}_1$ and $\mathcal{D}_2$. For example, this condition holds when the labels are generated by $y = f(G) + \epsilon \mathcal{N}(0, 1)$ for some 1-Lipschitz function $f$, where $\mathcal{N}(0, 1)$ is the standard normal distribution. We can obtain the size-generalization bound by applying the above theorem for the distribution of large graphs $\mathcal{G}_1$ and of small graphs $\mathcal{G}_2$. Thus, we only need to evaluate their Wasserstein distance. The Wasserstein distance can be large in the worst-case; thus, we here consider concrete examples of graph distributions.

First, we consider the case that undirected graphs are drawn from the *configuration model of d-regular graphs*. In this model, a graph is constructed by the following procedure: (1) It creates $N$ vertices with $d$ half-edges; (2) Then, it pairs the half-edges and connects them to obtain edges. We see that a learning problem on this distribution is size-generalizable.

**Proposition 14.** *Let* $\mathcal{G}$ *be a distribution of random d-regular graphs generated by the configuration model, and* $\mathcal{G}_{\leq N}$ *be the distribution conditioned on only graphs of size bounded by* $N$. *If* $N \geq (\log 1/\epsilon)^{\Omega(\log 1/\epsilon)}$ *then* $W(\mathcal{G}, \mathcal{G}_{\leq D}) = O(\epsilon)$.

This result can be generalized to a general distribution of graphs with large girth. Next, we consider the case where undirected graphs are drawn from a graphon. A *graphon* $\mathcal{W}$ is a function $\mathcal{W} : [0, 1] \times [0, 1] \to [0, 1]$. A graph $G_N$ is drawn from $\mathcal{W}$ if we first draw $N$ random numbers $x_1, \ldots, x_N \in [0, 1]$ uniformly randomly. Then, for each pairs $(x_i, x_j)$, we put an edge with probability $\mathcal{W}(x_i, x_j)$. This model extends Erdos–Renyi random graph and stochastic block model; see [37] for more detail.

**Proposition 15.** *Let* $\mathcal{W}$ *be a graphon. Let* $N_1$ *and* $N_2$ *be integers with* $N_1 < N_2$. *Let* $\mathcal{G}_{N_i}$ *be a distribution of graphs of* $N_i$ *vertices drawn from* $\mathcal{W}$. *If* $N_1 \geq 2^{O(1/\epsilon^2)}$ *then* $W(\mathcal{G}_{N_1}, \mathcal{G}_{N_2}) \leq \epsilon$.

Finally, we consider the case that $\mathcal{D}_N$ is obtained from $\mathcal{D}$ by the metric projection. Let $\Pi_N$ be the projection onto the space of graphs of size at most $N$, i.e., $\Pi(G) = \operatorname{argmin}_{G_N : |V(G_N)| \leq N} d(G, G_N)$.

**Proposition 16.** *Let* $\mathcal{G}$ *be any graph distribution and let* $\mathcal{G}_{\leq N} = \Pi(\mathcal{G})$ *be the projected distribution of graphs of size at most* $N$. *For any* $\epsilon > 0$, *there exists* $N$ *such that* $W(\mathcal{G}, \mathcal{G}_{\leq N}) \leq \epsilon$.

A drawback of this result is that an explicit bound of $N$ is not known, even for its deterministic variant in the bounded degree graphs (See Proposition 19.10 in [37]). The only known bound is for the bounded degree graphs with large girth [13].

**Comparison with Existing Results** Size-generalization of GNNs is reported on several tasks, but its theoretical analysis is limited. Yehudai et al. [72] studied the size-generalizability using the concept of $d$-pattern, which is information obtained from $d$-ball; it is similar to our $r$-profile. Their results are approximation-theoretic as they showed the (non-)existence of size-generalizable models but did not show how such models can be obtained by training on data. Xu et al. [71] proved the size-generalization of the max-degree function under several conditions on the training data and GNNs. Their result is essentially an approximation-theoretic as it assumes the dataset lies in and spans a certain space that is sufficient to identify the max-degree function.

## 6.4 Partially-Universal RBS-GNNs

Thus far, we assumed the universal GNNs are plugged into the RBS-GNNs for theoretical analysis. This assumption achieves the maximum expressive power in this framework; however, in practice, we often use expressive but more efficient GNNs such as GCN [30], GIN [70], or GAT [64]. Here, we discuss what will be changed if we made this modification.

Let $\equiv$ be an equivalence relation on graphs. We assume that $\equiv$ is *consistent with the weakly connected component decomposition*, i.e., if $G_1 \equiv H_1$ and $G_2 \equiv H_2$ then $G_1 + G_2 \equiv H_1 + H_2$, where the "+" symbol denotes the disjoint union of two graphs. We say that a function $h$ (resp. a randomized algorithm $\mathcal{A}$) is $\equiv$-*indistinguishable* if $h(G) = h(G')$ (resp. $\mathcal{A}(G) = \mathcal{A}(G')$ given the random sample) for all $G \equiv G'$. A GNN is $\equiv$-*universal* if it can learn any $\equiv$-indistinguishable functions. For example, it is known that GIN is universal with respect to the WL indistinguishable functions [70]. Let RBS-GNN[$\equiv$] be a class of RBS-GNNs that uses an $\equiv$-universal GNN $f$ in Equation (1). The following shows the partial universality of this architecture.

**Theorem 17.** *If $f$ is estimable and $\equiv$-indistinguishable, then it is estimable by an RBS-GNN[$\equiv$].*

One application of this theorem is extending the expressivity of GraphSAGE to the partial observation setting. GraphSAGE can represent the local clustering coefficient if we have the complete observation of the graph [22, Theorem 1]. We can prove the local clustering coefficient is estimable (Proposition 24 in Appendix D). By applying Theorem 17 to the equivalence relation $G_1 \equiv G_2$ defined by $f(G_1) = f(G_2)$ for all function $f$ representable by GraphSAGE, we obtain the following.

**Proposition 18.** *The mini-batch version of the GraphSAGE (Algorithm 2 in [22]) can estimate the local clustering coefficient.*

**Comparison with Existing Studies** Equivalence relations associated with GNNs are mainly studied in the context of the "limitation" of GNNs: If a GNN is $\equiv$-indistinguishable, then it cannot learn any function $h$ that is non $\equiv$-indistinguishable. Morris et al. [46] proved a message-passing type GNN cannot distinguish two graphs having the same $d$-patterns. Garg et al. [15] identified indistinguishable graphs of several GNNs, including GCN [30], GIN [70], GPNGNN [58], and DimeNet [32].

On the other hand, we should use non-universal GNNs in practice because more expressive GNNs have higher computational costs (e.g., universal GNNs [27] is more costly than the graph isomorphism test). We here considered RBS-GNN[$\equiv$] because $\equiv$-universal GNNs are theoretically tractable classes of non-universal GNNs. With similar motivation, [51] proposed GNNs parameterized by information aggregation pattern and proved the $\equiv$-universality, where $\equiv$ is induced by the aggregation patterns.

## 7 Conclusion

We answered the question "*What graph functions are representable by GNNs when we can only observe random neighborhoods?*" by proving the functions representable by RBS-GNNs coincides with the estimable functions in the random neighborhood model, which is equivalent to the uniformly continuous functions in the randomized Benjamini–Schramm topology. The word "(Some)" in our title was meant to emphasize the restriction to continuous graph functions. The result holds without any assumption on the input graphs, such as the boundedness. This result gives us a "functional analysis view" of graph learning problems and leads to several new learning-theoretic results. The weakness of our result is the poor dependency on the number of training instances, which is the trade-off for generality. We believe addressing this issue will be an interesting future direction. Another future direction links to the asymptotic behaviors of RBS-GNNs. Motivating example includes the motif counting problem, in which Bhattacharya et al. [5] proved the Horvitz–Thompson estimator is consistent and asymptotically normal. A corresponding result for the RBS-GNN will allow us to obtain a confidence interval of an estimation and to perform statistical testing.

**Potential Impact** Our work contributes an understanding of general graph learning models whose inputs are random samples of arbitrarily large graphs. Due to the theoretical nature of our results, we believe there will not be a direct nor indirect negative societal impact.

**Acknowledgement** We would like to thank the anonymous reviewers and the area chairs for their thoughtful comments which help us improve our manuscript. HN is partially supported by the Japanese Government MEXT SGU Scholarship No. 205144.

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
