# A Complete proofs

## A.1 Theorem 4

Theorem 4 states the universality of the proposed RBS-GNN. This theorem was proposed to address the question "Even if we have a powerful GNN, what kind of graph functions can we learn if the input graphs are too large to be computed as a whole?" posed in the Introduction. We think of this theorem from two perspectives. In one view, this theorem extends the universality of "complete-observation" GNNs in to "partial-observation". In another, the theorem reduced the universality of GNNs to only universal on estimable functions. This section presents proofs leading up to Theorem 4.

*Proof of Lemma 5.* We construct a canonical estimator from the original estimator. Let $\mathcal{E}$ be the original estimator and let $N$ be the total number of queries of to achieve accuracy $\epsilon^2/2$. We first construct an estimator $\mathcal{E}_1$. $\mathcal{E}_1$ samples an $N$ random balls using $\mathsf{RandomBallSample}(G, N, N)$ and simulates $\mathcal{E}$ on $\mathcal{E}_1$ using a permutation $\pi$ over the vertices of $B_1 \cup \cdots \cup B_r$. Because of the simulation, we obtain

$$\mathrm{Prob}_{S,\pi}\left[|f(G) - \mathcal{E}_1(G \mid S, \pi)| > \epsilon^2/2\right] < \epsilon^2/2. \tag{6}$$

Then, we construct the final estimator $\mathcal{E}_2$. $\mathcal{E}_2$ returns the expected value of the output of $\mathcal{E}_1$ over all simulations. Here,

$$\mathbb{E}_S[|f(G) - \mathcal{E}_2(G \mid S)]|] = \mathbb{E}_S[|f(G) - \mathbb{E}_\pi[\mathcal{E}_1(G \mid S, \pi)]|] \tag{7}$$
$$\leq \mathbb{E}_{S,\pi}[|f(G) - \mathcal{E}_1(G \mid S, \pi)|] \tag{8}$$
$$\leq \epsilon^2. \tag{9}$$

Thus, by the Markov inequality,

$$\mathrm{Prob}_S[|f(G) - \mathcal{E}_2(G \mid S)]| > \epsilon] \leq \epsilon. \tag{10}$$

$\square$

Using Lemma 5, we obtain the proof for Theorem 4.

*Proof of Theorem 4.* The output of the canonical estimator is determined by the isomorphism class of the subgraph induced by the balls. Hence, it is determined by the isomorphism classes of the weakly connected components of the induced subgraph. This means that we can write the canonical estimator as a function from the set of weakly-connected graphs: $h(\{C_1, \ldots, C_l\})$. Here, we can injectively map each $C_j$ as a finite-dimensional vector $z_j$ using a universal neural network $f$ since it has a size bounded by a constant (depending on $N$). Also, the number of connected components is bounded by a constant (depending on $N$). Thus, we can identify the function $h$ as a permutation-invariant function with a constant number of arguments whose inputs are finite-dimensional vectors. Therefore, we can apply Theorem 9 in [75], which shows that there exists continuous functions $g$ and $\phi$ such that $h(z_1, \ldots, z_m) = g(\sum_i \rho(z_m))$. for all $z_1, \ldots, z_m$. Because $\phi$ is approximated by a neural network, we can combine it with the GNN $f$; therefore, we obtain the proof. $\square$

## A.2 Theorem 7

Theorem 7 is perhaps the most important contribution of this work. This theorem continues to analyze the concept of estimable graph functions by providing a topology in which estimable functions are continuous and vice versa. We find that the result is quite useful when we want to apply functional analysis techniques to analyze graph learning problems. This section presents the proofs of Lemma 8 and Theorem 7.

*Proof of Lemma 8.* This is a variant of [37, Proposition 19.10], which is for a different topology (different computational model). We can prove our lemma by the same strategy, but here we provide a proof for completeness.

Let $r = \lceil \log 2/\epsilon \rceil$. We choose a maximal set of graphs $H_1, \ldots, H_N$ such that for all $i \neq j$, $d_{TV}(z_s(H_i), z_s(H_j)) > \epsilon/4$ holds on some $s \leq r$. We can see that such a set exists (see below). By

the maximality, for any graph $G$, there exists $j$ such that $d_{TV}(z_s(G), z_s(H_j)) \leq \epsilon/4$ for all $s \leq r$, which implies $d(G, H) \leq (1/2)^r + \epsilon/2 \leq \epsilon$.

We show the upper bound of $C$. In the proof, we represent $G$ by an $r$-tuple $(z_1(G), \ldots, z_r(G))$ of probability distributions, where each $z_s(G)$ lies on $2^{s^s \times s^s} = 2^{(\log 1/\epsilon)^{O(\log 1/\epsilon)}}$-dimensional simplex for $s \leq r$. Because the packing number of $d$-dimensional simplex in the total variation distance (equivalently in the $l_1$ metric) is $(1/\epsilon)^{O(d)}$, we cannot choose more than $2^{2^{(\log 1/\epsilon)^{O(\log 1/\epsilon)}}}$ points whose pairwise distance is at least $\epsilon/4$. $\square$

*Proof of Theorem 7.* Suppose $f$ is estimable. For any $\epsilon > 0$, we choose a canonical estimator $\mathcal{A}$ of accuracy $\epsilon$. Then we have $|f(G) - f(H)| \leq |f(G) - \mathcal{A}(G)| + |\mathcal{A}(G) - \mathcal{A}(H)| + |\mathcal{A}(H) - f(H)|$. Here, the first and last terms are at most $\epsilon$ by the definition of $\mathcal{A}$ with probability at least $1 - \epsilon$, respectively. We take $\delta = 2^{-r}\epsilon$ for the second term. Then, for any $G, H$ with $d(G, H) \leq \delta$, we have $d_{TV}(z_r(G), z_r(H)) \leq \epsilon$; hence, by the optimal coupling theorem,[2] there exists a coupling between $Z_r(G)$ and $Z_r(H)$ such that $P(Z_r(G) \neq Z_r(H)) = d_{TV}(z_r(G), z_r(H)) \leq \epsilon$. Thus, the output of the algorithm $\mathcal{A}$ coincides on $G$ and $H$ with a probability at least $1 - \epsilon$. Therefore we have $|f(G) - f(H)| \leq 3\epsilon$ with probability at least $1 - 3\epsilon$. By taking the expectation, we obtain the result.

Suppose $f$ is uniformly continuous. For any $\epsilon > 0$, let $\delta > 0$ be the corresponding constant in the continuity definition. Take a $\delta/2$-net $\{H_1, \ldots, H_C\}$ and let $r = \lceil \log 4/\delta \rceil$. The algorithm $\mathcal{A}$ performs random sapling to estimate the distribution $(z_1(G), \ldots, z_r(G))$ with accuracy $\delta/2$ with probability at least $1 - \epsilon$. Then, it outputs $f(H_j)$, where $H_j$ is the nearest neighborhood of $G$. By the construction, the algorithm finds $H_j$ with $d(G, H_j) \leq \delta$ with probability at least $1 - \epsilon$. Therefore, we have $|f(G) - \mathcal{A}(G)| = |f(G) - f(H_j)| \leq \epsilon$ with probability as least $1 - \epsilon$. $\square$

It should be emphasized that the space of all graphs equipped with the randomized Benjamini–Schramm topology is *not* compact, because there is a continuous but not uniformly continuous function; the average degree function is such an example.

## A.3 Proofs for Applications

This section provides the proofs for the theoretical applications section in the main part (Section 6). Most notably, the proofs for Theorem 10, 11, and 13 are provided here.

*Proof of Proposition 9.* Let $M$ be the endpoints of the random edges. Then, $M$ induces a uniform distribution on the vertices of $G$. Any run with $Z_r(G) \cap M = \emptyset$ can be coupled with $Z_r(G')$; so the coupling probability is

$$P(M \cap Z_r(G) = \emptyset) = \sum_{x \in M} P(x \notin Z_r(G)) \tag{11}$$

$$\leq |M| r^r / n \tag{12}$$

$$= 2r^r \delta. \tag{13}$$

By the optimal coupling theorem, we have $d_{TV}(z_r(G), z_r(G')) = 2r^r\delta$. Therefore,

$$d(G, G') = \sum_{r=1}^{\infty} 2^{-r} d_{TV}(G_r, G'_r) \tag{14}$$

$$\leq \sum_{r=1}^{\infty} 2^{-r} \min\{1, 2r^r\delta\} \tag{15}$$

$$\leq s^s \delta + 2^{-s} \tag{16}$$

for any $s$. By putting $s = \log\log(1/\delta)$, we obtain the result. $\square$

If $M$ is chosen adversarially, we cannot obtain the inequality (12). In particular, if $M$ contains a vertex $x$ with a large PageRank, as the probability of $x \in Z_r(G)$ is large, we cannot bound the distance.

---

[2]See [7], or `pages.uoregon.edu/dlevin/AMS_shortcourse/ams_coupling.pdf` (May, 2021).

*Proof of Theorem 10.* We use the following inequality that bounds the Rademacher complexity by the covering number $C_{\mathcal{F}}(\epsilon)$ of the function space $\mathcal{F}$:

$$R_n(\mathcal{F}) \leq \inf_{\epsilon > 0} \left\{ \epsilon + O\left( \sqrt{\frac{\log C_{\mathcal{F}}(\epsilon)}{n}} \right) \right\}. \tag{17}$$

We choose an $\epsilon/2$-net of the graphs of size $C(\epsilon/2)$ by Lemma 8. Then, we define an (external) $\epsilon$-cover of the space of 1-Lipschitz functions by the piecewise constant functions whose values are discretized by $\epsilon/2$, where the pieces are the Voronoi regions of the $\epsilon$-net; it is easy to verify this is an $\epsilon$-cover of the space of 1-Lipschitz functions. This shows $C_{\mathcal{F}}(\epsilon) \leq (2/\epsilon)^{C(\epsilon/2)} = 2^{2^{2^{O(\log(1/\epsilon) \log \log(1/\epsilon))}}}$. By substituting $\epsilon$ satisfying $O(\log(1/\epsilon) \log \log(1/\epsilon)) = \log \log \log(n/\log n))$, we obtain the result.

$\log C_{\mathcal{F}}(\epsilon) = 2^{2^{2^{O(\log 1/\epsilon \log \log 1/\epsilon)}}}$. We set $\epsilon$ to be $O(\log 1/\epsilon \log \log 1/\epsilon) = \log \log \log(n/\log n)^2$. Then, by definition, $\sqrt{\log C_{\mathcal{F}}(\epsilon)/n} = 1/\log n$.

We try to evaluate $\epsilon$. We see $\epsilon$ satisfies $\log 1/\epsilon \log \log 1/\epsilon = \Omega(\log \log \log n)$.

Recall that $x \log x = y$ iff $y = e^{W(x)}$ where $W(x)$ is the Lambert W function. Since $W(x) = \log(x/\log x) + \Theta(\log \log x / \log x)$, we have $y \geq x/\log x$. By using this formula, we have $\log 1/\epsilon = \Omega(\log \log \log n / \log \log \log \log n)$. $\square$

*Proof of Theorem 11.* By Theorem 7, an estimable function $f$ is uniformly continuous. Let $\delta$ be the constant for $\epsilon$ for the continuity. By Lemma 8, there is an $\delta$-net $\{H_1, \ldots, H_C\}$ and let $N(\delta)$ be the maximum number of vertices in the graphs in the $\delta$-net. Our algorithm $\mathcal{A}$ outputs $\mathcal{A}(G) = f(H_j)$ where $H_j$ is the nearest neighbor of $G$. This algorithm achieves the accuracy of $\epsilon$ because $d(G, H_j) \leq \delta$. Also, the algorithm can be constructed only accessing graphs of size at most $N = \max_j |V(H_j)|$. Hence, $f$ is size-generalizable. $\square$

*Proof of Lemma 12.* This is an adaptation of [62] to our metric space. Their proof only uses the "easy" direction of the Kantorovich–Rubinstein duality, which holds on any metric space. Hence, we obtain this lemma.

To be self-contained, we will give a proof. For any 1-Lipschitz function $f$ and any coupling $\pi$ between $\mathcal{D}_1$ and $\mathcal{D}_2$, we have the following "easy" direction of the Kantorovich–Rubinstein duality:

$$\mathbb{E}_1[f(G_1)] - \mathbb{E}_2[f(G_2)] = \mathbb{E}_{(G_1, G_1) \sim \pi} \mathbb{E}[f(G_1) - f(G_2)] \tag{18}$$

$$\leq \mathbb{E}_{(G_1, G_2) \sim \pi} \mathbb{E}[d(G_1, G_2)] \tag{19}$$

$$\leq W(\mathcal{G}_1, \mathcal{G}_2). \tag{20}$$

By putting $f = (h - h')/2$, we obtain

$$\mathbb{E}_1 |h(G) - h'(G)| - \mathbb{E}_2 |h(G) - h'(G)| \leq 2W(\mathcal{G}_1, \mathcal{G}_2). \tag{21}$$

Hence,

$$\mathbb{E}_1 |y - h(G)| \leq \mathbb{E}_1 |y - h'(G)| + \mathbb{E}_1 |h(G) - h'(G)| \tag{22}$$

$$= \mathbb{E}_1 |y - h'(G)| + \mathbb{E}_1 [h(G) - h'(G)] + \mathbb{E}_2 |h(G) - h'(G)| - \mathbb{E}_2 |h(G) - h'(G)| \tag{23}$$

$$\leq \mathbb{E}_1 |y - h'(G)| + \mathbb{E}_2 |h(G) - h'(G)| + 2W(\mathcal{G}_1, \mathcal{G}_2) \tag{24}$$

$$\leq \mathbb{E}_2 |y - h(G)| + \mathbb{E}_1 |y - h'(G)| + \mathbb{E}_2 |h(G) - h'(G)| + 2W(\mathcal{G}_1, \mathcal{G}_2). \tag{25}$$

By taking the infimum over $h'$, we obtain the theorem. $\square$

*Proof of Theorem 13.* We obtain the result by combining Theorem 10 and Lemma 12. $\square$

*Proof of Proposition 14.* Let $\mathcal{G}_N$ be the distribution of random $d$-regulra graphs of size $N$. Then, we can see that $Z_r(G_N) \mid G_N \sim \mathcal{G}_N$ has no cycle with probability at least $1 - r^{O(r)}/N$. This implies that we can couple $Z_r(G) \mid G \sim \mathcal{G}$ and $Z_r(G_{\leq N}) \mid G_{\leq N} \sim \mathcal{G}_{\leq N}$ with probability at least $1 - r^{O(r)}/N$. By putting $r = \log 1/\epsilon$, we obtain the result. $\square$

*Proof of Proposition 15.* This follows from the proof of Lemma 10.31 and Exercise 10.31 in [37]. □

*Proof of Proposition 16.* We can choose $N$ by the maximum number of vertices in the $\epsilon$-net. Then, we have $d(G, \Pi(G)) \leq \epsilon$. Thus the Wasserstein distance is bounded by $\epsilon$. □

*Proof of Theorem 17.* We introduce a helper concept, $\equiv$-*indistinguishably estimable*, which is a class of functions that is estimable by $\equiv$-indistinguishable computation on $r$-profile. By the same argument as Theorem 4, we can show that RBS-GNN[$\equiv$] can represent $\equiv$-indistinguishably estimable function.

Now we prove that if a function $f$ is estimable and $\equiv$-indistinguishable, then it is $\equiv$-indistinguishably estimable. Because $f$ is estimable, there exists $\delta > 0$ such that $|f(G) - f(H)| \leq \epsilon/2$ if $d(G, H) < \delta$. We fix a $\delta$-net $\{H_1, \ldots, H_C\}$ of the graph space. We consider a quotient space of the graphs by $\equiv$, select a representative $[G]$ to each quotient, and assign $H_i$ to $[G]$ which is the nearest neighbor of the representative $G$.

Our estimator $\mathcal{A}$ is the following. First, we obtain a subgraph $S$ by sampling sufficiently many vertices to be $d(S, G) \leq \delta$ with probability at least $1 - \epsilon$. Second, we take the representative $[S]$ of the equivalent class containing $S$. Finally, we output the value $f(H)$, where $H$ is the nearest neighbor of $[S]$. By construction, $\mathcal{A}$ is an $\equiv$-indistinguishable computation after the sampling. Here,

$$|f(G) - \mathcal{A}(G)| \leq |f(G) - f(S)| + |f(S) - f([S])| + |f([S]) - f(H)| \leq \epsilon \quad (26)$$

with probability at least $1 - \epsilon$, where the first term in the right-hand side is at most $\epsilon/2$ with probability at least $1 - \epsilon$ due to the sampling and continuity, the second term is zero due to the $\equiv$-indistinguishability, and the last term is at most $\epsilon/2$ due to the $\delta$-net and uniform continuity. □

# B  Extended Results: Finite-Dimensional Vertex Features

We can extend our framework to the vertex-featured case. We assume the vertex features are in $[0, 1]^d$. This assumption is well-aligned with the pre-processing step in practice where vertex features are normalized [30, 50, 51].

The estimability is defined similarly, where we additionally assume that the estimation is uniformly continuous with respect to the vertex features on the sampled subgraph (in the standard topology of $\mathbb{R}^d$). Then, we can prove the RBS-GNN can estimate arbitrary estimable vertex-featured graph parameters.

The difficulty is how to define the topology on the vertex-featured graphs. As in the non-featured case, We want to define $Z_r(G)$ by the "frequency" of the graphs. However, since there are uncountably many vertex-featured graphs, we need a technique. To address this issue, we fix an $\epsilon$-net on $[0, 1]^d$; it has the cardinality of $(1/\epsilon)^d$. Then, we approximate the vertex features of the sampled graph by the elements of the $\epsilon$-net by the uniform continuity. Then, the number of "vertex-featured graphs" of $N$ vertices is bounded by $((1/\epsilon)^d)^{O(N \times N)}$; hence we can define the randomized Benjamini–Schramm topology. The space is totally bounded since the $\epsilon$-net is constructed by combining the $\epsilon$-net of the graph and the $\epsilon$-net of $[0, 1]^d$. Note that this makes no significant difference on the size of the $\epsilon$-net since the difference is absorbed in the nested power.

# C  Extended Results: Vertex Classification Problems

In this section, we extend our framework for the graph classification problem to the vertex classification problem.

The vertex classification problem is usually defined as follows. We are given a set of graphs $G_1, \ldots, G_N$ with the "supervised vertices" $S_1 \subseteq V(G_1), \ldots, S_N \subseteq V(G_N)$ and the labels $y_u$ on the supervised vertices. The task is to find an equivariant function $h \colon G \mapsto (y_1, \ldots, y_n) \in \mathcal{Y}^{V(G)}$. This formulation, however, is not suitable for large graphs because it needs to output values to all the vertices. Here, we recognize a vertex classification problem as a rooted graph classification problem as in [50]: The input of the problem is a set of pairs $(y_{v_i}, (G_i, v_i))$ of rooted graphs $(G_i, v_i)$ and the label $y_{v_i}$. The goal is to find a function $h$ such that $h((G, v)) \approx y_v$. It should be noted that a graph $G$

with a supervised nodes $S \subseteq V(G)$ in the original formulation is transformed to $|S|$ rooted graphs $\{(G, v) : v \in S\}$.

Our framework for the graph classification problem is easily extended to the rooted graph classification problem. We first modify our computational model by assuming the root of the graph is available at the beginning of the computation. Then, the estimability of the function is defined in the same way using this computational model. Then, we modify the RBS-GNN to have one additional ball centered at the root vertex, i.e.,

$$\text{RBS-GNN}((G, v)) = g\left(f(C_0), \sum_{j=1}^{N_C} f(C_j)\right) \tag{27}$$

where $B_0, \ldots, B_k$ are the random balls obtained by RandomBallSample where the root of $B_0$ is conditioned by $v$, and $C_0, \ldots, C_{CS}$ are the weakly connected components of $G[B_0 \cup \cdots \cup B_k]$ where $C_0$ contains $v$. We can prove that any estimable vertex parameter in the random neighborhood model for the rooted graph is estimable using the RBS-GNN. Also, by extending the randomized Benjamini–Schramm topology to the rooted graphs, we see the estimability coincides with the uniform continuity in the randomized Benjamini–Schramm topology of rooted graphs.

Using this topology, we can obtain the vertex classification version of the results in Section 6. As the number of rooted graphs of $N$ vertices is $N$ times larger than the number of non-rooted graphs of $N$ vertices, the covering number of the rooted graph space is larger than that of the non-rooted graph space. But this makes no significant difference because this gap is absorbed in the nested logarithm.

This formulation gives several consequences on the vertex classification problem.

- We can evaluate the required number of supervised vertices to obtain the desired accuracy. In this formulation, each supervised vertex $v$ corresponds to a rooted graph $(G, v)$. Thus, if the supervised vertices are chosen randomly, the required number of supervised vertices is evaluated by the Rademacher complexity of the model. In particular, we can obtain a model with an accuracy $\epsilon$ from the constantly many supervised vertices.

- We can characterize the difficulty of a vertex classification problem with different supervision using transfer learning. Imagine a situation that the supervised vertices have large degrees, but we want to predict the vertex property on low-degree vertices. This situation can be recognized that the training and test rooted graph distributions, $\mathcal{G}_{\text{train}}$ and $\mathcal{G}_{\text{test}}$, are different. Therefore, we can apply Lemma 12 to obtain an estimation of the test error, which involves the Wasserstein distance of these distributions.

## D    Extended Results: Practical Applications

Many real-world graph parameters are estimable in our framework. This section provides a review of graph parameters and their estimability in our random neighborhood model. Most notably, this section demonstrates the usage of Theorem 7 for practical graph parameters and provide the proof for Proposition 18. In addition, we provide experimental results on real-world datasets to verify our theoretical claims.

### D.1    Graph Parameters

For convenience, we re-state the random neighborhood model and the *estimable* definitions here. The original definitions were provided in Section 3.2.

**Definition 19** (Random Neighborhood Model). *The random neighborhood computational model allows the following three queries given an input graph $G$:*

- SampleVertex($G$): Sample a vertex $u \in V$ uniformly randomly.
- SampleNeighbor($G$, $u$): Sample a vertex $v$ from the neighborhood of $u$ uniformly randomly, where $u$ is an already obtained vertex.
- IsAdjacent($G$, $u$, $v$): Return whether the vertices $u$ and $v$ are adjacent, where $u$ and $v$ are already obtained vertices.

*This computational model induces an estimability definition and a topology, named random Benjamini-Schramm, on the graph space $\mathcal{G}$.*

**Definition 20** (Constant-Time Estimable Graph Parameter). *A graph parameter $p$ is constant-time estimable on the random neighborhood model (estimable for short) if for any $\epsilon > 0$ there exists an integer $N$ and a randomized algorithm $\mathcal{A}$ in the random neighborhood model such that $\mathcal{A}$ performs at most $N$ queries and $|\mathcal{A}(G) - p(G)| < \epsilon$ with probability at least $1 - \epsilon$ for all graphs $G \in \mathcal{G}$.*

### D.1.1 Non-estimable Graph Parameters

We first see that an estimable parameter is bounded as follows.

**Proposition 21.** *An estimable parameter $p$ is bounded.*

*Proof.* Recall that an estimable parameter is uniformly continuous (Theorem 7). Let $\delta$ be the constant for $\epsilon = 1$ for the continuity. Let us fix an $\delta$-net $\{G_1, G_2, \dots\}$ of the graph space using the totally boundedness of the space (Lemma 8). Then, $p$ is bounded by $C = 1 + \max_i p(G_i)$. In fact, for any graph $G \in \mathcal{G}$, we have

$$p(G) \leq |p(G) - p(G_i)| + |p(G_i)| \leq C \tag{28}$$

where $G_i$ is the nearest neighbor of $G$ in the $\epsilon$-net. $\qquad\square$

Example 2 in Section 3.2 states that the number of vertices and the min/max degrees are not estimable; these immediately follow form Proposition 21 since they are unbounded parameters. Similarly, the average degree is unbounded so it is not estimable. The connectivity function is an example of bounded but non-estimable graph parameter.

**Proposition 22.** $p(G) = 1[G \text{ is connected}]$ *is not estimable.*

*Proof.* We prove this proposition by giving a counter example showing $p$ violates the definition for continuity. Since continuity is equivalent to estimability, such counter example would also disprove the estimability of $p$.

We first fix $\epsilon = 1/2$. Then, we choose two graphs $G_1$ and $G_2$ such that $G_1$ is the disjoint union of two cliques of size $N$ and $G_2$ is obtained from $G_1$ by adding one edge between them. The figure below demonstrates for the case $N = 6$.

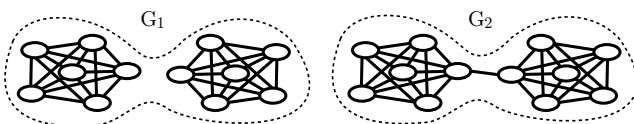

We see that $d(G_1, G_2) \leq \delta$ for any chosen $\delta > 0$ if $N$ is sufficiently large. This is because the distribution $z_r(G_1)$ and $z_r(G_2)$ of isomorphism classes only different at the event that one of the two connected vertices of $G_2$ is sampled. The probability for such event becomes increasingly insignificant when $N$ is sufficiently large. Hence, the distance $d(G_1, G_2)$ can be arbitrarily small as stated above. However, by the definition of the connectivity function, for any $N$ we always have $|p(G_1) - p(G_2)| = 1 > 1/2$. Hence, $p$ is not continuous, and by Theorem 7, it is also not estimable. $\qquad\square$

### D.1.2 Estimable Graph Parameters

The following propositions proves statements in Example 3.

**Proposition 23.** *The triangle density is a uniformly continuous parameter and estimable.*

Using Theorem 7, it is clear that we only need to prove estimability or continuity. We show both proofs for this case as a demonstration. For simplicity, we assume the input graph $G$ is undirected.

*Proof for Estimability.* We show that the triangle density is estimable by constructing a random algorithm and prove that this algorithm estimates the triangle density to an arbitrary precision dependent only on the number of random samples (Definition 1). The randomized algorithm can be implemented under the random neighborhood computational model (Definition 19).

---

**Algorithm 2** Triangle Density Estimation in the Random Neighborhood Model

---

1: **procedure** ISTRIANGLE($G, u, v, q$)
2:     $uv \leftarrow$ IsAdjacent($G, u, v$);
3:     $uq \leftarrow$ IsAdjacent($G, u, q$);
4:     $qv \leftarrow$ IsAdjacent($G, q, v$);
5:     **return** $uv \wedge uq \wedge qv$;                    $\triangleright \wedge$ is the logical "and".
6: **procedure** TRIANGLEDENSITY($G, T$)
7:     triangles $\leftarrow 0$;
8:     **for** $i$ in $1, \dots, T$ **do**
9:         $u, v, q \leftarrow$ SampleVertex($G$);              $\triangleright$ Runs 3 times to get 3 samples.
10:        triangles $\leftarrow$ triangles $+$ IsTriangle($G, u, v, q$);
11:    **return** $\frac{1}{T}$triangles;

---

The procedure IsTriangle in Algorithm 2 return 1 if the three input vertices induce a triangle and 0 otherwise. Let $X$ be the output of IsTriangle given three random vertices $u, v,$ and $q$ from graph $G$, and $\bar{X}$ be the output of TriangleDensity. By definition, the expectation $\mathbb{E}(X)$ is the true triangle density $p_\Delta$. Since the $p_\Delta$ and $\text{Var}(X)$ are clearly finite, we can apply the Chebyshev's concentration bound to the sample average $\bar{X}$ with $T$ samples to obtain the following. For any $\epsilon > 0$,

$$\mathbb{P}(|\bar{X} - p_\Delta)| \geq \epsilon) \leq \frac{\text{Var}(X)}{\epsilon^2 T}. \tag{29}$$

This bound shows that if we take $T = O(\epsilon^{-3})$ samples, then with probability at least $1 - \epsilon$ we obtain an estimation less than $\epsilon$ from the true value. Note that $T$ is only dependent on the precision $\epsilon$ and not the size of $G$; this shows the intuition behind the constant-time nature of our RBS-GNN.

$\square$

*Proof for Continuity.* We now prove that the triangle density $p_\Delta$ is uniformly continuous in the randomized Benjamini-Schramm topology. For a given $\epsilon > 0$, we choose $\delta = 2^3\epsilon$. Let $G_1$ and $G_2$ be two graphs satisfying $d(G_1, G_2) \leq \delta$. We denote two random variable $X_1$ and $X_2$ to represent the event that random sampling from $G_1$ and $G_2$ obtained a triangle. $X_1$ and $X_2$ follows $z_\Delta(G_1)$ and $z_\Delta(G_2)$ distributions, respectively. By the optimal coupling theorem, the random sampling on $G_1$ and $G_2$ can be coupled with probability at least $1 - \epsilon$.

$$d_{TV}(z_\Delta(G_1), z_\Delta(G_2)) = \min_{(X_1, X_2)-\text{couplings}} \mathbb{P}(X_1 \neq X_2) \tag{30}$$

Hence, by the definition of the triangle density, these differs at most $\epsilon$.                    $\square$

Using a similar technique, we can prove the estimability or equivalently uniformly continuity of the local clustering coefficient.

**Proposition 24.** *The local clustering coefficient is uniformly continuous and estimable.*

### D.2 Graph Classification in Random Neighborhood Model

We show the results for RBS-GNN on social networks datasets in the TUDatasets repository [47]: COLLAB, REDDIT-BINARY, and REDDIT-MULTI5K. We preprocess these datasets in the same way as proposed by Xu et al. [70]. Because of this pre-processing, each vertex has a feature vector representing its position in the degree distribution. The reason for such setting is because in 1-WL, the degree determines the initial coloring [48]. A summary of the datasets is given in Table 1.

To simulate the random ball sampling procedure of RBS-GNN, we pre-sample the original datasets and use these random samples in both training and testing. Table 1 shows the sampling setting we used to report the results in Table 2. We prepared multiple other settings for $r$, $b$, and $k$, see the provided source code for more detail (`supp/notebooks/Preprocessing.ipynb`). Let $N_G(\cdot)$ be the neighborhood function of the sampled input graph, we construct a $K$-layers $f$ as follows.

$$h_G^{(\ell)}(u) = \text{MLP}^{(\ell)} \left( \sum_{v \in N_G(u)} h^{(\ell-1)}(v) \right), \ell = 1, \dots, K \tag{31}$$

Table 1: Overview of the graph classification datasets. This is a small part of the TUDataset [47]. $|\mathcal{G}|$ denotes the total number of graphs in the dataset, $\overline{\nu}(G)$ denotes the average number of nodes per graph, $|c|$ denotes the number of classes, $d$ denotes the dimensionality of vertex features (created by [70]). $r$ denotes the radius of random balls and also the branching factor. $k$ denotes the number of random balls. $\%$ denotes the average coverage of random balls in terms of the number of edges. $\%$ Memory denotes the relative data storage size.

| DATASETS | $|\mathcal{G}|$ | $\overline{\nu}(G)$ | $|c|$ | $d$ | $r$ | $b$ | $k$ | $\%|E(G)|$ | $\%$ Memory |
|---|---|---|---|---|---|---|---|---|---|
| COLLAB | 5000 | 74.5 | 3 | 367 | 2 | 5 | 3 | $58.3 \pm 22.9$ | 55.7 |
| RDT-BINARY | 2000 | 429.6 | 2 | 566 | 3 | 5 | 3 | $37.3 \pm 28.6$ | 14.9 |
| RDT-MULTI5K | 5000 | 508.5 | 5 | 734 | 3 | 5 | 3 | $23.8 \pm 17.7$ | 14.2 |

$$f(G) = \sum_{\ell=1,\ldots,K} \sum_{u \in G} h_G^{(\ell)}(u), \tag{32}$$

where $\text{MLP}^\ell$ is a single layer MLP with no activation. Let $g$ be a 2-layers MLP with ReLU activation functions, the final output of RBS-GNN$[\equiv_2]$ is given similar to Equation (1):

$$\text{RBS-GNN}[\equiv_2](G) = g\left(\sum_j f(G)\right). \tag{33}$$

The implemented RBS-GNN is denoted by RBS-GNN$[\equiv_2]$ because its GNN component $f$ resembles a message-passing GNN such as GIN [70]. In all our experiments, the graph neural network $f$ has 4 propagation layers and 5 MLP layers, each of these layers have 32 ReLU hidden units (see `supp/src/rbsgnn/models/mpgnn_batched.py` and `supp/notebooks/Cross-Validation Scores.ipynb`). Regularization methods are weight decay ($10^{-3}$), learning rate decay (initialized at 0.01, step size 50, $\gamma = 0.5$), and dropout (0.5).

**Computational Resources** We run all our experiments on a single computer having a single Intel CPU (i7-8700K3.70GHz), 64GB DDR4 memory, and a NVIDIA GeForce GTX 1080Ti GPU with CUDA 11.3 (driver version 465.31). The system runs Linux Kernel 5.12.6. Our model's prototype is implemented using Python 3.9 and PyTorch 1.8.1+cu111 (see `supp/src/requirements.txt` for the detail of the Python environment).

**Cross-Validation Scores** Reporting the 10-folds (also 3-folds and 5-folds) cross-validation scores for graph learning model is a common task in the literature [11, 69, 70]. We compare our practical implementation of RBS-GNN to existing benchmarks in terms of 10-folds cross validation scores. We show the results for other baselines reported by Xu et al. [70]. These baselines includes WL-subtree, PatchySan, and AWL (see Section 7 in [70] for more detail). Note that RBS-GNN only has access to partial inputs for both training and testing procedures. The fractions of observed edges and storage memory are shown in Table 1.

Table 2: Best test accuracy (in percentage) for the graph classification task. Note that RBS-GNN only has access to 20~60 percent of edges and 15~50 percent of node features (Table 1).

| MODELS | COLLAB | RDT-BINARY | RDT-MULTI5K |
|---|---|---|---|
| GIN-0 | $80.2 \pm 1.9$ | $92.4 \pm 2.5$ | $57.5 \pm 1.5$ |
| WL subtree | $78.9 \pm 1.9$ | $81.0 \pm 3.1$ | $52.5 \pm 2.1$ |
| PatchySan | $72.6 \pm 2.2$ | $86.3 \pm 1.6$ | $49.1 \pm 0.7$ |
| AWL | $73.9 \pm 1.9$ | $87.9 \pm 2.5$ | $54.7 \pm 2.9$ |
| RBS-GNN$[\equiv_2]$ | $80.3 \pm 1.5$ | $79.0 \pm 1.9$ | $44.0 \pm 1.4$ |

The result in Table 2 shows that at best we can achieve similar result for COLLAB while observing only 55.7% of the data. Note that this experiment is different from any random pooling or drop-out techniques because we use random balls for *both* train and test. The results for REDDIT datasets

are also quite similar to the results of complete-observation models. As shown in Table 1, we only observe about 14% of the REDDIT original datasets. We selected such extreme example to show that although by a small observation, in some cases GNNs can still predict well. This observation implies that the true labeling function of these dataset is smooth in the random Benjamini-Schramm topology.

**Size Generalization** Our Theorem 11 identified size-generalizability with estimability. We verify this theoretical result experimentally as follows. For each dataset, we split into two sets of train and test data (0.5/0.5 split). Furthermore, the train set consists of smaller graphs while the test set has larger ones. We then report the test accuracy for targets being the global clustering coefficient (estimable) and the max-degree (non-estimable). In machine learning, it is often more beneficial to work with classification rather than regression; therefore, we categorized the clustering coefficients and the max-degree values into 5 classes. Each class represents a 20-percentile proportion of the data (see `supp/notebooks/preprocessing.ipynb`). The test accuracy is reported in Table 3.

Table 3: Test accuracy (in percentage) of RBS-GNN[$\equiv_2$] for the size-generalization task.

| Global Clustering | | | Max-Degree | | |
|---|---|---|---|---|---|
| COLLAB | RDT-BINARY | RDT-MULTI5K | COLLAB | RDT-BINARY | RDT-MULTI5K |
| $23.7 \pm 3.7$ | $33.7 \pm 1.1$ | $40.0 \pm 2.0$ | $16.1 \pm 2.1$ | $43.4 \pm 1.2$ | $39.8 \pm 3.1$ |