# OpenReview forum: "Learning on Random Balls is Sufficient for Estimating (Some) Graph Parameters"
_NeurIPS.cc/2021/Conference — NeurIPS 2021 Poster_

### Official Review · Reviewer_2yke · 2021-07-07

**Rating:** 7
**Confidence:** 3

**Summary:**

To take the words of the authors directly, this paper answers the question "What graph functions are representable by GNNs when we can only observe random neighborhoods?" In answering this question, the authors consider a notion of an estimable function in a random neighborhood model, which they then show is equivalent to uniform continuity in a metric topology, which they dub the randomized Benjamini-Schramm topology. With this perspective in hand, they show that sufficiently powerful graph neural networks coupled with universal approximators are capable of estimating uniformly continuous functions in this topology. They then discuss implications of this in terms of robustness, sampling complexity, and generalization.

**Limitations And Societal Impact:**

Limitations are addressed in body of the paper, societal impact is not an issue for this work.

**Main Review:**

I will begin by discussing the strengths of this paper, then move on to the weaknesses, followed by some brief comments.

**Strengths**

1. This paper is very well-presented. Although its contents are technically demanding, the results are presented in such a way that lends to easy understanding. Despite this paper being somewhat outside of my main research area, I found it to be not too difficult to digest. The authors should be applauded for refraining from using extremely fancy machinery to convey their results, which is all-too-tempting in papers like this. Because of this, I think this paper is of broad interest to the graph learning community.

2. The approach taken by this work is quite general. As the authors point out, it makes essentially no assumption on the graph structure itself, only demanding that the function to be estimated is estimable under their sampling model. Related to this is the simplicity of their approach: although they study estimation under a specific random ball sampling algorithm, the RBS-GNN is not particularly exotic. That is to say, the results presented are apparently applicable in many contexts, as discussed in Section 6.4, where weaker classes of GNNs are considered.

3. The problem studied is quite relevant: there is great interest in applying graph neural networks approaches to huge graphs, with many approaches dedicated to ways to split graphs up or sample them in clever ways to avoid storing a humongous data structure. This paper offers a nice set of results for this particular problem, which will be useful to practitioners in understanding what constitutes an "estimable" parameter and what sampling strategies should work.

4. The proposed random Benjamini-Schramm topology seems to be novel.

**Weaknesses**

1. Although the authors admit this, the generality of their approach leads to some issues with bounds having poor dependance on the graph size and estimation accuracy. This is addressed briefly in a few places, but it would have been nice to see some more concrete examples regarding this. For the benefit of the research community, this addition would ground this work in practice better.

3. The main paper does not show any empirical validation of the ideas in the paper. I do understand that this is primarily a theoretical paper, but the fact that a specific sampling method was studied motivates the inclusion of some empirical results. I did see that there are some in the supplementary material, which is appreciated, but these were somewhat weak in my opinion. The RBS-GNN clearly underperformed relative to competing approaches, which is understandable given the limited observation scheme. However, I would have liked to see a comparison with other neural approaches that sample subgraphs for the purpose of embedding.

4. Despite my aforementioned appreciation of the presentation of this work, there are a few things that could use a bit more explanation. I list these below.

i. Section 5.2. When defining the sampling distance, a few things stick out to me. First, the explanation of an $r$-profile is unclear: I am not sure what this distribution is meant to look like. I see that there is some randomness that comes up due to the random ball sampling, but how this translates to a random vector is not explained very well. Given that the metric topology of this sampling distance is the core contribution of this paper, this warrants more effort in explanation. I would have suggested moving part of Section 6 to the supplementary material in order to make room for this.

ii. Again in Section 5.2, eq (2). It isn't too surprising that this is a reasonable distance to choose, casting aside my previous doubts about $r$-profiles. However, this could use a bit more explanation. Reading it, the weighting by $2^{-r}$ treats smaller sampling radii as more important, while downweighting larger sampling radii: and of course, this kind of weighted sum of a difference of sequences is a common example of a distance between sequences in functional analysis. In this context, this could be interpreted as saying that the distance is small for graphs that exhibit similar motif structures, but perhaps with differing global structure. Could you please explain this in more detail? Is the sampling distance the only appropriate distance for defining this topology, or is it one in some class of appropriate metrics for this task?

**Comments**

1. Overall thoughts: I found this paper to be quite enjoyable and informative. The presentation is quite good, and the results are novel, providing an interesting perspective on randomized approaches to graph parameter estimation. I appreciated how the authors tied the notion of an estimable graph parameter to a novel metric topology, which allowd them to approach this problem from a functional analysis perspective. Overall, despite the issues I pointed out on practicality and some didactic concerns, I think this is a fine paper, and should be accepted.

**Editorial** (No impact on decision)

1. Line 228: "Benjamini-Schram" is misspelled.

2. Line 303: "numbere" is misspelled.

3. Some of the citations do not look very good. Please take another pass on these, **with a particular focus on capitalization.** Some problematic citations I noticed were [20],[31],[34],[40],[42],[43],[56],[58],[61],[63/64],[71],[72]. There are also inconsistencies on venue citation, for instance between [13/15] and [11/21]. I would suggest using the `@string` macro in BibTeX to ensure consistent venue naming, e.g.,

```
@string{iclr={International Conference on Learning Representations}}

@article{
title={Absurdly Deep Convolutional Neural Networks},
author={John Smith},
journal=iclr,
}
```


**Time Spent Reviewing:**

6

---

> ### Author Response · Authors · 2021-08-07
> **We address your concerns numbered 3i and 3ii**
>
> We truly appreciate your detailed review of our paper and your editorial suggestions are also extremely helpful to us.
> We are going to add more experimental evidence to our paper; we understand that experimental results would make our results more approachable.
> We also revised Section 5 as your comment number 3 in the "weakness" section suggested.
> As we generally agree with your criticism and aim to improve with weakness point 1 (elaborate on the sample complexity) and 2 (add experimental results) in the future version.
> Here, we would like to provide a quick summary of our response to comments (3i) and (3ii) in this discussion.
> ---
> > 3i. [...]  explanation of an r-profile [...]
>
> The r-profile $Z_r(G)$ can be thought of as a random vector in which each dimension corresponds to an isomorphism class of graphs sampled from $G$.
> To simplify, let us consider there is one root ($k = 1$) and radius $r = 2$.
> In this case, $Z_2(G)$ consists of two elements: graph of two isolated vertices and graph of two vertices with one edge.
> $Z_2(G)$ can be modeled as a two-dimensional (indicator) vector: $[0, 1]$ represents the event two isolated vertices are sampled and $[1, 0]$ represented the event one edge is sampled.
> Now it's clear that $Z_r(G)$ is a random vector whose "randomness" comes from SampleVertex and SampleNeighbor.
> Therefore, if we take the expectation over these two factors, we get $z_r(G) = \mathbb{E}(Z_r(G))$, which is the distribution of isomorphism classes on the input graph $G$.
> Let's say $G$ is a graph with four vertices and one edge (it looks like this: o o o--o); then the distribution $z_2(G)$ can be written as $[0.5, 0.5]$.
> Following your comments, we will add these explanations to the future version of our manuscript.
> ---
> > 3ii. [...] Eq (2) in 5.2 [...]
>
> Your understanding is correct and we will update our manuscript to explain the intuition behind Equation (2) better.
> Such definition also reveals the local-to-global structure of the distance between two input graphs, which is more interesting than the continuity notion on standard topology (based on the distance of adjacency matrices).
> Similar to our discussion with reviewer TQV6, basically, Eq. (2) has an exponential discount factor to make the distance bounded; note that the factor $2$ is not important and can be replaced to any constant greater than $1$.
> Intuitively, as you pointed out, Eq. (2) formally describes the idea: "two graphs are similar" if "samples from these graphs look similar"'.
> If the number of samples is fixed, the total variation distance ($d_{TV}(\cdot, \cdot)$) can be used to represent this idea.
> To aggregate all the total variation distances for all numbers of samples, we put the discount factor $2^{-r}$ and sum them up to obtain our distance.
> Note that this definition comes from the definition of the Benjamini--Schramm topology, which is also defined by the sum of the total variation distances multiplied by $2^{-r}$, where the definition of the ``samples'' differs from ours (theirs are only dependent on the random root vertex while ours depends on both random root vertices and random neighbor sampling).
> ---
> We hope this response cleared some of your concerns. Please feel free to let us know if you have further follow-up questions in the rolling discussion.

---

### Official Review · Reviewer_eHqG · 2021-07-14

**Rating:** 5
**Confidence:** 2

**Summary:**

The paper uses sublinear algorithms to estimate certain graph functions.

The paper is essentially a property testing (sublinear algorithms) paper posing as a GNN paper. This is not necessarily a bad thing if the connection is strong enough; however, this connection does not become very clear to me.

The main relation to GNN is very vague as core concepts such as universality are not well-defined. I presume the universal approximation theorem is meant here, but I cannot be sure.


**Limitations And Societal Impact:**

Fine

**Main Review:**

In section 5.1 an important restriction is mentioned that somehow does not become clear in Theorem 3.
(the theorem is only applicable to the continuous functions in  the randomized Benjamini–Schramm topology)

There are some applications mentioned, but they are not very illuminating. Could you perhaps state some concrete examples of functions that can be estimated?

Not clear what this means:
A rooted graph (G, v) is a graph G augmented with a vertex v in V (G)

WeaklyConnectedComponents quadratic query complexity? Needs to check all possible adjacency matrix entries? What if the degrees are super-constant? Does that mean that the sampling is also super-constant?

The proof idea of Theorem 3 feels unfinished. Why is good to be able to list all isomorphism classes of all graphs on x vertices?

Shouldn't Theorem 3 mention sampling complexity in some way or another?

**Time Spent Reviewing:**

3

---

> ### Author Response · Authors · 2021-08-07
> **We believe there is some fundamental misunderstanding**
>
> We believe there is a fundamental misunderstanding here because this paper does not "use sublinear algorithms to estimate graph functions", but technically speaking our analyzes show that sampling is a method to convert a standard algorithm (e.g. a GNN) to a sublinear algorithm.
> Furthermore, as we clarified in lines 74-75, we use "GNN" because theoretically, GNNs can be universal approximators for all graph functions.
> However, as explained in lines 75-76, our discussion here applies to all graph learning methods, as the title and the abstract suggested.
> About the universality notion, universality means the existence of the universal approximation theorem.
> This is a common terminology (e.g, lines 20-24 and Section 6.4, or [this paper](http://proceedings.mlr.press/v97/maron19a.html).
> The following addresses your main reviews.
> ---
> > In section 5.1 an important restriction is mentioned that somehow does not become clear in Theorem 3. (the theorem is only applicable to the continuous functions in the randomized Benjamini–Schramm topology)
>
> Theorem 3 stated that estimable functions in the random neighborhood model can be estimated by the theoretical RBS-GNN model.
> Then, in Section 5.2, the connection between estimable functions and continuous functions in Benjamini-Schramm topology is established with Theorem 5 (estimable iff continuous).
> Therefore, it is trivial to see that Theorem 3 included the restriction you mentioned.
> ---
> > There are some applications mentioned, but they are not very illuminating. Could you perhaps state some concrete examples of functions that can be estimated?
>
> In lines 153-154, we provided concrete examples of estimable and non-estimable functions (proofs for these examples are provided in Appendix D.1.1 and D.1.2).
> Also, Proposition 16 proved that the local clustering coefficient is estimable even with 1-WL universal models.
> ---
> > Not clear what this means: A rooted graph (G, v) is a graph G augmented with a vertex v in V (G).
>
> This defines a "rooted graph" (see, e.g. [wikipedia](https://en.wikipedia.org/wiki/Rooted_graph)).
> We will update the definition as "$(G, v)$ is a graph $G$ with a distinguished vertex $v \in V(G)$'' if it caused confusion.
> ---
> > WeaklyConnectedComponents quadratic query complexity?
> Needs to check all possible adjacency matrix entries?
> What if the degrees are super-constant?
> Does that mean that the sampling is also super-constant?
>
> Let $C$ be the number of sampled vertices;
> it is a constant in the sense that it is independent of the graph size ($|V(G)|$ or $|E(G)|$).
> As we assumed the sampling oracle (Section 3.2, neighbor sampling is constant time), the complexity of the sampling depends on the number of samples $C$.
> WCC is applied to the sampled vertices so the complexity is quadratic in $C$ (say, $O(C^2)$) but is still a constant.
> ---
> > The proof idea of Theorem 3 feels unfinished.
> Why is good to be able to list all isomorphism classes of all graphs on x vertices?
>
> We checked our proof sketch for Theorem 3 to see if there is an editorial mistake, we think the provided summary is quite enough given the space limit.
> We refer interested readers to our Appendix for the complete proof of Theorem 3.
> About listing all isomorphism classes, it's good to be able to list all isomorphism classes because we can adapt the topology used in the Benjamini-Schramm convergence, which was developed for bounded degree graphs.
> For bounded degree graphs, given a radius $r$, there are only a finite number of isomorphism classes.
> This simple property leads to the fact that the bounded degree space is totally bounded and the notion of convergence can be discussed.
> To extend such topology to general graphs, we propose to analyze the random neighborhood sampling.
> Since we have a constant number of neighbors for each node after sampling, we can also enumerate all isomorphism classes given radius $r$ like in the classical Benjanimi-Schramm framework; however, the twist here is that we can't estimate any graph parameters but only continuous parameters in the proposed RBS topology.
> ---
> > Shouldn't Theorem 3 mention sampling complexity in some way or another?
>
> No, this is essentially a universality theorem for the space of estimable graph functions.
> Theorem 3 describes the property of the function space (i.e., RBS-GNN is a universal approximator of continuous functions).
> It only states the existence of universal approximators.
> Please kindly refer to our introduction and related work sections, we have cited extensively the literature on the universality theorem.
> Furthermore, as we assumed the sampling oracle, neighborhood sampling runs in constant time.
> In case you meant sample complexity (in the machine learning sense), we provided the Rademacher complexity of this function class in Section 6.2.
> ---
>
> Please let us know if our answers cleared your concerns. We hope they help you to view our work in a more positive light as well as to focus on our contributions.

---

> > ### Comment · Reviewer_eHqG · 2021-09-10
> > **Ack**
> >
> > I acknowledge that I have read the rebuttal. When I did so, I increased my rating (16 Aug).

---

### Official Review · Reviewer_VmDC · 2021-07-16

**Rating:** 5
**Confidence:** 4

**Summary:**

The paper proposes to use randomly sampled subgraphs from a large graph to estimate the function value computed from the entire graph. The paper has done extensive theoretical analysis using the "randomized Benjamini–Schramm topology". The model also has analyzed learnability issues when estimating these functions.

**Ethical Concerns:**

No ethical concerns were detected.

**Limitations And Societal Impact:**

No potential negative societal impact is known.

**Main Review:**

Strengths

+The main problem addressed in this paper is the estimation of graph functions through sampling neighborhoods on the graph. This paper seems to be the first theoretical analysis of this problem. This problem is related to two problems. The first one is the estimation of a value of a large graph (e.g. the average degree) through sampling, which is a traditional problem. The second problem is the analysis of GNNs generalizing to different sizes of graphs.

+Using the "randomized Benjamini–Schramm topology" brings a new technique to the area is another contribution of the submission.

Weaknesses

-The problem studied in this paper is unrealistic. The introduction says the motivation is the learning problem where the input is a huge graph and the output is a single value. At least I don't know such a problem exists: we don't have thousands of social networks and need to classify them. The experiment in the appendix does not use any data supporting the motivation. Actually, the experiment samples tiny graphs, all of which does not have computation or memory issues.


-The function to be estimated is unknown, so we are not sure whether it is estimable. Particularly, the learned model as a function is not necessarily estimable. For example, we can use the learning model to estimate the maximum degree. Therefore, it is unknown where the analysis is applicable.

-By reading this submission, I have a feeling that it overly claims the contribution. For example, the title put parenthesis over "some". Do you mean "some" can be removed? Also, line 270:  "... guarantees the asymptotic convergence on any graph learning problem without assuming any graph structure." I believe "any graph learning problem" is not true: it is limited to the estimable functions. There are also other cases.

-There should be a more thorough introduction of the literature. There is plenty of work on randomized testing graph properties or computing graph parameters. Not all of them are based on "randomized Benjamini–Schramm topology" -- why this topology is advantageous?

Nevertheless, I still think the research carries value, but the authors may want to find the scenario for the analysis. For example, are there any learning problems that have prior knowledge that the unknown function is estimable?



**Time Spent Reviewing:**

2

---

> ### Author Response · Authors · 2021-08-07
> **We explain our motivation and address the weaknesses**
>
> We appreciate your time reviewing our paper and we think that your concerns are logical.
> We believe there is some slight misunderstanding of our motivation to propose the Randomized Benjamini-Schramm topology.
> Our motivation is that many existing works employ sampling (esp. large-scale settings), so we propose the RBS topology to characterize exactly which function can be approximated.
> Since random sampling is pervasive in practical graph learning systems, our RBS topology is a useful tool analysis tool.
> Here, we address the weaknesses you mentioned by providing some context and details.
> ---
> > The problem studied in this paper is unrealistic. [...]
>
> The point cloud classification problem using GNNs is such an example.
> As we mentioned in the Related Work section (lines 58-59), there is some success in posing the point cloud classification problem as a graph classification problem [1].
> Also, in our Introduction section (line 35-36), our analysis is valid for the vertex classification problem (Appendix C) and there are plenty of works that use sampling methods to solve the problem (lines 54-61).
>
> The point here is that applications of GNNs are mainly social networks but there are many other problems that can be posed as large networks.
> We would further argue that the _learning problem where the input is a huge graph and the output is a single value_ has many existing instances (mentioned above) as well as potential: Input is a large social network of a country, the output is the predicted infection rate; input is a large network of communication conditioned on some hashtag, the output can be if this kind of communication is fake news or not.
> Indeed, these datasets are difficult to collect and maintain, but it does not mean the general problem is unrealistic.
> ---
> > The function to be estimated is unknown, so we are not sure whether it is estimable. [...] (also [...] are there any learning problems that have prior knowledge that the unknown function is estimable?)
>
> While the function to be estimated is unknown, existing methods already apply random sampling in their learning process for regularization or scalability reasons.
> Motivated by this practice, we propose our topology as an analysis tool applicable exactly when the inputs are partial observation.
> Therefore, this topology is *not* artificial, but it is a natural choice to analyze such a setting.
> To elaborate on your comment about the max-degree function, it is true that this function is not estimable and yet existing models can learn as well as size-generalize it (under some assumptions).
> Even the paper discussed the conditions for size-generalizability of GNNs on the max-degree function still has to know what function to estimate (they *know* it is the max-degree function).
>
> Generally speaking, making assumptions about the unknown function in machine learning is difficult (see e.g., [this paper](http://www.econ.upf.edu/~lugosi/mlss_slt.pdf)).
> Therefore, the common practice is to make "powerful models" with some inductive bias.
> So, as pointed out in the introduction, we would like to focus on the question: What kind of function can we estimate when the inputs are random samples?
> Our results contribute to the fact that random sampling on GNNs (at best) can only learn estimable or RBS-continuous graph functions.
> On the max-degree example, our theory gives a somewhat trivial result that any graph learning model with partial observation cannot learn the max-degree function because this function is not estimable in the random neighborhood model.
> This gives a clear characterization for models like GraphSAGE and to the best of our knowledge, this is the first analysis tool to do so.
>
> We believe that your main point here is how our continuity assumption (continuity in the randomized BS topology) is reasonable.
> We give an answer by providing 3 arguments: (a) Compare our continuity notion with the standard continuity in other works, (b) Highlight the importance of understanding random sampling models in contrast with existing regularization techniques in computer vision, and finally (c) our continuity assumption is perhaps unavoidable in large-scale analyses.
>
> (a) In fact, existing universality and functional analyses of graph learning methods also assume continuity in the standard global topology [2,3,4,5].
> Our distance given by Equation (2) can be understood as a decomposition from local (small $r$) to global (large $r$) distances,
> which is in contrast to the standard topology that only takes the global structure into account.
> Such a definition is beneficial for analyzing constructive methods where the model builds global structure from local structural inputs (e.g. GraphSAGE and modern GNN), which is the motivation of this study.
>
> (b) In recent literature on learning methods for graphs, many regularization methods are proposed (DropEdge, DropNodes, data augmentation, etc.).
> These methods often have the same motivation rooted in computer vision literature (i.e. based on dropout method)
> While it is well-assumed in CV that some artificial perturbation should not change the target of the input, the same thing can't be said when inputs are graphs.
> In this context, our continuity notion is useful for characterizing models in which the input graph is a random sample of the original graphs (e.g. GraphSAGE).
>
> (c) Since only a limited number of organizations have access to large networks as well as the computing power to process them, we believe theoretical studies of very large-scale problems can bring insights.
> In such large-scale settings, assuming the input graph is just a random sample is not only realistic but also unavoidable.
> This is where our continuity assumption finds itself better than the standard continuity assumptions.
> ---
> > By reading this submission, I have a feeling that it overly claims the contribution. [...]
>
> The word "some" in the title refers to the set of estimable parameters (also the set of continuous graph functions in the Benjamini-Schramm topology).
> This word can be removed and the sentence is still logically correct; however, we believe this word emphasizes that our theory shows exactly when the statement is true.
> Removing "some" from our title would be an overt claim of contribution, so we put "some" there to emphasize our contribution that we characterize what graph function is estimable under random sampling.
>
> Our wording in line 270 needs some context, we are referring to the learning problems on the general graph space, and Theorem 8 above this statement clearly stated the 1-Lipschitz assumption.
> As we review the sentence, we feel the word "any graph learning problem" is indeed somewhat misleading if taken out of context and will update it to provide more clarity.
> ---
> > There should be a more thorough introduction of the literature.
> There is plenty of work on randomized testing graph properties or computing graph parameters.
> Not all of them are based on "randomized Benjamini–Schramm topology" -- why this topology is advantageous?
>
> Our motivation is to analyze the neighborhood sampling technique, which is commonly used in practice.
> We choose this topology as it naturally fits the technique.
> We also note that we did provide an overall review of the literature in Sections 2 and 3, especially lines 131 to 137.
> ---
>
> We hope these answers addressed your concerns and help to correct some misunderstandings of our work.
> Please let us know your thoughts and if you want us to further elaborate on any point.
>
> ---
> Note: All these citations appeared in our manuscript.
>
> [1] Diego Valsesia, Giulia Fracastoro, and Enrico Magli. Learning localized generative models for 3d-point clouds via graph convolution.International Conference on Learning Representations, 2019
> [2]Nicolas Keriven and Gabriel Peyré. Universal invariant and equivariant graph neural networks.arXiv preprint arXiv:1905.04943, 2019.
> [3]Nicolas Keriven, Alberto Bietti, and Samuel Vaiter. Convergence and stability of graph convolutional networks on large random graphs.arXiv preprint arXiv:2006.01868, 2020.
> [4]Haggai Maron, Ethan Fetaya, Nimrod Segol, and Yaron Lipman. On the universality of invariant networks.arXiv preprint arXiv:1901.09342, 2019.
> [5]Hoang NT and Takanori Maehara. Graph homomorphism convolution. In Proceeding of the 37th International Conference on Machine Learning, pages 7306–7316. PMLR, 2020.

---

> > ### Comment · Reviewer_VmDC · 2021-08-23
> > **Thank you for further explanation**
> >
> > Here are some further comments:
> >
> > 1. "Indeed, these datasets are difficult to collect and maintain, but it does not mean the general problem is unrealistic." -- I think at least it is fair to say that the research will not impact many people since not many people will have a problem addressed by the research.
> >
> > 2. About my statement of assuming the underlying function is estimable. I would argue most functions that are interesting, particularly in the context of GNN learning problems, are not estimable. This is different as "making assumptions about the unknown function in machine learning is difficult ". For example, in an image classification problem, we don't know the form of the function, but we know that the function is likely NOT linear. In the case of graph learning, I believe the most interesting functions are somewhat related to the size of the graph and thus are not estimable. Using your example, the size of a hashtag network is likely to play a role in an interesting function (e.g. influence of the hashtag) over the network. An estimable function may have to be independent of the size of the graph (maybe other quantities as well). I don't know how realistic this assumption is.
> >
> > 3. About the title: I would encourage the author to find a better word to show exactly the scope of graph parameters studied by this research. In my view, estimable graph functions only take a tiny fraction of overall graph functions.

---

> > > ### Author Response · Authors · 2021-08-24
> > > **Thank you for the additional comments**
> > >
> > > We truly appreciate you for spending your time to interact with us.
> > >
> > > *Reply to (1)*: Now we see that you have a different opinion than ours on this point: we believe having applications in point-cloud application and large-scale vertex classification problems are enough to claim that our result have impact on research community.
> > > Graph sampling techniques are also widely used in practice but they lack theoretical understanding --- which our work contributed.
> > > On this point, we are happy to hear other reviewers' opinions.
> > >
> > > *Reply to (3)*: We agree with your opinion here.
> > > However, please understand that universal models (e.g. tensorized GNN) assume the input space is "bounded", which means even the existing "universal models" also cover just a "tiny fraction" of overall graph functions (any size input).
> > > Moreover, these bounded functions are included in our estimable functions (by setting $r$ sufficiently large) so our class of functions are **strictly larger than the existing "universal" class** despite sounding restrictive.
> > > We hope you can see this point.
> > >
> > > *Technically, our framework shows the sampling can extend the universal functions in the bounded space to the universal functions in the unbounded space with the new topology.*
> > >
> > > We would like to discuss your 2nd point in more detail.
> > >
> > > ---
> > > > "I would argue most functions that are interesting, particularly in the context of GNN learning problems, are not estimable. [...] For example, in an image classification problem, we don't know the form of the function, but we know that the function is likely NOT linear. [...] I believe the most interesting functions are somewhat related to the size of the graph and thus are not estimable."
> > >
> > > We disagree with this point because (a) continuity reflects the intuition that adding/removing a small number of vertices does not change the result, and (b) even the target function is not estimable, our theoretical framework is still useful.
> > > Let us elaborate on each point.
> > >
> > > (a) We argue that there will be several interesting estimable functions in GNN context.
> > > The estimability (i.e., the continuity in the randomized Benjamini-Schramm topology) reflects the intuition that adding/removing a small number of vertices/edges does not change the result (i.e., it's a *continuity in the graph size direction*, see e.g., Proposition 7).
> > > There are several examples that will likely satisfy this property.
> > > For example, the functions in the pose estimation problem (in point cloud) and the classification problems of social or web graphs will be estimable.
> > > In your example of the influence of the hashtag, we can define the function of interest by the fraction of the influence instead of the size of the influence.
> > > Then the obtained function will likely be estimable, and will still be useful in many applications.
> > >
> > > (b) Our theory is useful even though the interesting function is not estimable.
> > > As you said, if we know the function is not linear, we will avoid training a linear model.
> > > Our theory provides a corresponding decision in GNN context: **if the function of interest is not estimable, we should avoid training a sampling-based models such as GraphSAGE**.
> > > For example, again in your hashtag example, if we actually need the size of the influence (instead of the fraction), our theory tells us to avoid the sampling-based method or to decompose the problem into estimating the fraction and estimating the size of the graph itself; here, our theory again indicates that the former can be solved by a sampling-based GNN whereas the latter cannot be.
> > > Conversely, if in practice we design a model which uses random-sampling from graphs, our theory shows that the model has an implicit bias called "estimable functions" or "continuous in RBS-topology".
> > >
> > > ---
> > >
> > > In conclusion, many large-scale models take subgraph sampling for granted without knowing the implicit bias, our work contribute a theoretical understanding of such practice.
> > > We provides a framework to study and discuss graph functions, which is missing in the current literature.
> > > All in all, we are grateful you have spent time to discuss with us..
> > >
> > > ---
> > >
> > > P/S: On a more lighthearted note, this very discussion is an example of what we want to add to the landscape of graph learning.
> > > We can clearly discuss which graph functions are interesting or not under a rigorous mathematical framework.
> > > Such discussion might be rather difficult under the current theoretical framework (as we mentioned above).

---

### Official Review · Reviewer_TQV6 · 2021-07-17

**Rating:** 6
**Confidence:** 3

**Summary:**

This paper studies a theoretical framework to address learning problems over graphs under partial observation constraints. In this context, a Random Balls Sampling GNN architecture based on random neighborhood model is introduced and its expressibility and learnability are characterized theoretically. Applications of the proposed architecture to settings with perturbations and size-generalizability are also discussed.

**Main Review:**

1. The notion of estimability of a graph parameter is central to the theoretical claims in the paper. Therefore, I recommend that the authors elaborate more on this aspect in Section 3.2. Moving Definition 18 and some of the discussions in Appendix D.1 to the main paper will greatly improve the readability of this section of the paper.

2. Similarly, the notion of 'uniformly continuous' graph parameter must be explicitly defined in Section 5.2.

3. What is the intuition behind the factor $2^{-r}$ in equation (2)?

4. In Section 6.3.1, what does the notation $G_{\leq N}$ signify? Also, shouldn't you have $|p(G) - {\cal A}(G_{\leq N})|\leq \epsilon$ in the definition of a size generalizable parameter $p$ for the algorithm ${\cal A}$ that operates on the given dataset?

5. The main paper lacks experimental evaluations and limited experiments are relegated to the Appendix. It would be useful to have an experimental setup in the main paper to illustrate the applicability of various theoretical claims in the paper.

The theory in this paper is tangential to my expertise and therefore, I am not fully familiar with other theoretical works in this context. However, the notions of learning from partially observed graphs and size generalizability are interesting and would be of interest to a wider audience. The readability of the paper could be improved by some reorganization of the content between the main paper and Appendices and including formally defining the notions of estimability, uniform continuity, size generalizability etc.



**Time Spent Reviewing:**

3

---

> ### Author Response · Authors · 2021-08-07
> **We address your comment points 3-5**
>
> We are grateful for your time and your structured comments. We have revisited our manuscript and made some changes with your comments 1 and 2 in mind. Our discussion here will attempt to provide some answers to your review points 3 to 5.
>
> ---
> > 3. What is the intuition behind the factor $2^{-r}$ in equation (2)?
>
> Basically, this is just an exponential discount factor to make the distance bounded; the factor $2$ is not important and can be replaced to any constant greater than $1$.
> Intuitively, we wanted to define "two graphs are similar" if "samples from these graphs look similar"'.
> If the number of samples is fixed, the total variation distance ($d_{TV}(\cdot, \cdot)$) can be used to represent this idea.
> To aggregate all the total variation distances for all numbers of samples, we put the discount factor $2^{-r}$ and sum them up to obtain our distance.
> Note that this definition comes from the definition of the Benjamini--Schramm topology, which is also defined by the sum of the total variation distances multiplied by $2^{-r}$, where the definition of the ``samples'' differs from ours (theirs are only dependent on the random root vertex while ours depends on both random root vertices and random neighbor sampling).
> ---
> > 4. In Section 6.3.1, what does the notation $G_{\leq N}$ signify?
> Also, shouldn't you have $|p(G) - {\cal A}(G_{\leq N})|\leq \epsilon$ in the definition of a size generalizable parameter for the algorithm that operates on the given dataset?
>
> $G_{\leq N}$ (line 296) represents a graph drawn from a distribution of graphs of size bounded by $N$.
> Hence, there is no direct correspondence between $G \in \mathcal{G}$ and $ G\_{\le N} \in \mathcal{G}\_{\le N}$.
> We can use another notation like $G'$ or $H$ instead of $G_{\le N}$, but as we wanted to clarify the graphs are bounded, we used this notation.
> ---
> > 5. The main paper lacks experimental evaluations and limited experiments are relegated to the Appendix. It would be useful to have an experimental setup in the main paper to illustrate the applicability of various theoretical claims in the paper.
>
> We are adding more experimental results in the final version.
> Although our intention is to focus on the theoretical results, we can see that it would be more illustrative to have more experimental results.
> ---
>
> We hope these answers addressed your concerns. Please let us know if you would like us to elaborate more on any point.

---

### Author Response · Authors · 2021-08-16
**We look forward to having a discussion**

Dear reviewers,

First, we would like to express our gratitude for your time and your comments on our work.
Since some of you had a misunderstanding of the paper (as we addressed them in our rebuttal on August 8th), we would like to resolve these misunderstandings in the discussion phase.
It would be greatly appreciated if you can kindly let us know your opinions after reading our rebuttal.
Thank you for your time and we look forward to having a more in-depth discussion with you.

---

### Decision · Program_Chairs · 2021-09-27

**Decision:**

Accept (Poster)

**Comment:**

This paper considers estimation of a real number associated to a large graph. Some examples of quantities that one might wish to estimate are the density of triangles (or relatedly the clustering coefficient). Rather than observing the entire graph, it is assumed that algorithms have access to a specific random sampling procedure whereby random nodes are selected and then their neighborhoods are randomly explored. A variation on the Benjamini-Schramm topology is introduced and the set of estimable graph properties is characterized in terms of continuity in this topology. A number of other results are presented regarding generalization across multiple graph sizes, etc.

The proposed sampling/observation model is reasonable and placed within context of related observation models. While the work is entirely theoretical, its results clearly demarcate what is possible or impossible with regards to the specific sampling model. Thus, there are natural implications for what can be learned by graph neural networks. The work seems likely to spur further investigation with other sampling models and to make tighter connections to finite size graph neural networks.